# Early Zinc Supplementation Enhances Epididymal Sperm Glycosylation, Endocrine Activity, and Antioxidant Activity in Rats Exposed to Cadmium

**DOI:** 10.3390/ijms26104589

**Published:** 2025-05-10

**Authors:** Sergio Marín de Jesús, Rosa María Vigueras-Villaseñor, Edith Cortés-Barberena, Joel Hernández-Rodríguez, Sonia Guadalupe Pérez-Aguirre, Sergio Montes, Leticia Carrizales-Yáñez, Isabel Arrieta-Cruz, Marcela Arteaga-Silva

**Affiliations:** 1Doctorado en Ciencias Biológicas y de la Salud, Universidad Autónoma Metropolitana-Iztapalapa, Av. Ferrocarril San Rafael Atlixco 186, Col. Leyes de Reforma 1ª. Sección, Alcaldía Iztapalapa, Ciudad de México C.P. 09340, Mexico; uamarindejesusergio@gmail.com (S.M.d.J.); soniagaguirre@gmail.com (S.G.P.-A.); 2Laboratorio de Biología de la Reproducción, Instituto Nacional de Pediatría, Av. Insurgentes Sur 3700-Letra C, Insurgentes Cuicuilco, Coyoacán, Ciudad de México C.P. 04530, Mexico; rmvigueras@yahoo.com.mx; 3Departamento de Ciencias de la Salud, División de Ciencias Biológicas y de la Salud de la Universidad Autónoma Metropolitana-Iztapalapa, Av. Ferrocarril San Rafael Atlixco 186, Col. Leyes de Reforma 1ª. Sección, Alcaldía Iztapalapa, Ciudad de México C.P. 09340, Mexico; cobe@xanum.uam.mx; 4Cuerpo Académico en Quiropráctica, Universidad Estatal del Valle de Ecatepec, Av. Central s/n, Esq. Leona Vicario, Col. Valle de Anáhuac, Secc. A, Ecatepec C.P. 55210, Mexico; joelhr19@hotmail.com; 5Unidad Académica Multidisciplinaria Reynosa-Aztlán, Universidad Autónoma de Tamaulipas, Calle 16 y Lago de Chapala, Col. Aztlán, Reynosa C.P. 88740, Mexico; montesergio@gmail.com; 6Coordinación de Innovación y Aplicación de la Ciencia y la Tecnología (CIACYT)-Facultad de Medicina, Universidad Autónoma de San Luis Potosí, San Luis Potosí C.P. 78210, Mexico; letcay@uaslp.mx; 7Departamento de Investigación Básica, Instituto Nacional de Geriatría, Secretaria de Salud, Ciudad de México C.P. 10200, Mexico; arrieta777@mail.com; 8Laboratorio de Neuroendocrinología Reproductiva, Departamento de Biología de la Reproducción, División de Ciencias Biológicas y de la Salud de la Universidad Autónoma Metropolitana-Iztapalapa, Av. Ferrocarril San Rafael Atlixco 186, Col. Leyes de Reforma 1ª. Sección, Alcaldía Iztapalapa, Ciudad de México C.P. 09340, Mexico

**Keywords:** sperm maturation, cadmium, glycosylation, zinc, spermatogenesis

## Abstract

Sperm maturation involves changes in plasma membrane glycosylation for fertilization. Cadmium (Cd) exerts a negative effect by disrupting testicular and epididymal function, altering antioxidant activity. Zinc (Zn) is an essential element known for its antioxidant properties, role in testosterone synthesis, and support of spermatogenesis. However, its effect on sperm membrane glycosylation, as well as endocrine and antioxidant activity, after exposure to Cd has remained unexplored. This study evaluated the impact of Zn on epididymal sperm glycosylation, endocrine activity, and antioxidant activity in Cd-exposed rats. Four groups of male Wistar rats were analyzed: control, Cd-exposed, Zn-supplemented, and Zn + Cd groups. On postnatal day 90, tissues and blood were collected for Zn and Cd quantification, testosterone levels, antioxidant activity, histological analysis, and sperm quality. The results showed that Cd concentration increased significantly, reduced testosterone levels, modified antioxidant activity, and caused structural damage in the epididymis. The Cd-exposed group showed disrupted glycosylation and distribution patterns and reduced sperm quality. The Zn + Cd group showed lower Cd accumulation, preserved testosterone levels, restored antioxidant activity, and preserved glycosylation patterns and sperm quality. This study highlights the protective role of Zn in mitigating Cd-induced reproductive toxicity, probably through the competitive inhibition of Cd uptake and antioxidant support, thereby preserving fertility.

## 1. Introduction

Male reproductive success depends on several interrelated physiological events, including hormonal regulation through the action of steroid hormones such as testosterone, and the activity of the organs of the reproductive system, particularly the processes that occur within the testis and epididymis. In particular, it can be pointed out that both compartments of the testis are highly relevant for fertility since it is in the Leydig cells of the interstitium that testosterone is produced; meanwhile, in the seminiferous tubules, spermatogenesis takes place through a complex and strictly regulated process that consists of the differentiation of germ cells (spermatogonia) to give rise to sperm [1,2]. At the end of this process and after spermiation, the morphologically differentiated sperm are released from the seminiferous tubules to be directed towards the rete testis, but they remain functionally immature; that is, they lack the potential to achieve progressive movement or capacitate themselves and are not optimal for recognizing and fertilizing an oocyte [2,3], so sperm motility and fertilization potential are post-testicular modifications acquired during their transit through the epididymis in the male reproductive tract, for which they need to be in contact with the fluids secreted via this organ [4,5]. These fluids provide the sperm with nutrients, maturation, and survival, which are important for fertilization [4,6], and protect them during transportation and storage [7,8]. Anatomically, the epididymis is divided into three regions: caput, corpus, and cauda, and in rodents, there is an initial segment [9]. It has a pseudostratified epithelium with different types of specialized cells [8,9,10], which are responsible for the process called epididymal sperm maturation [8,9], which consists of physiological and biochemical modifications of the sperm during its journey through the epididymis [1], such as the remodeling and assembly of glycan chains into glycoconjugates (glycoproteins and glycolipids) [11,12], added to the surface of sperm during spermatogenesis, so that the glycans present are further modified during their journey through the epididymis by the activity of glycosyltransferases and glycosidases in a process called glycosylation [11,13]. Among the glycans modified on the surface of sperm during their epididymal maturation, N-acetylglucosamine, sialic acid, and mannose stand out, as they are involved in the functions of recognition and interaction with the oocyte to perform the acrosomal reaction and fertilization [11,14].

On the other hand, it has been reported that, in recent decades, male fertility problems have increased and become relevant to health because they contribute significantly to the infertility of couples [15]. Among the causes that correlate with the dysfunction of the reproductive system and, therefore, with poor sperm quality [16] is exposure to environmental toxins [15,17], such as heavy metals, which cause adverse effects in the human population [18], especially cadmium (Cd), which is not essential for cellular processes but can accumulate in organs, especially the liver and kidneys [19]. The main form of contact with Cd is occupational exposure through the inhalation of industrial vapors and processes such as smelting and electroplating, battery manufacturing, alloying, welding, color pigments, and nuclear power plants, among others [20,21,22]. In terms of non-occupational exposure, the main sources are tobacco smoke and the consumption of contaminated food and water [21]. Exposure to Cd has been reported to cause various adverse health effects, including reduced fertility in humans and animals [23,24]. The presence of Cd in blood plasma and semen has been reported to be associated with azoospermia, oligospermia, and male infertility [25,26]. Likewise, Cd causes damage to the reproductive organs [27] in both subjects with environmental exposure and those with occupational exposure [28]. In rats, Cd can alter epididymal function and interfere with the glycosylation of the plasma membrane of sperm, thereby reducing male fertility [29]. Experimental evidence has confirmed the positive effect of dietary supplements such as antioxidants and vitamins against the damage caused by Cd; treatment with zinc (Zn) prevents the harmful effects of Cd [30,31,32]. Zn is an essential trace element, considered a micronutrient necessary for metabolism and one of the most important due to its properties and various biological functions [33,34], as it plays a fundamental role in physiological processes such as the regulation of the cell cycle, apoptosis, the organization and synthesis of DNA and RNA, proteins, lipids, and carbohydrates, as well as the stability of cell membranes [35,36]. The role of Zn in oxidative damage has been widely studied because it is an element with inert antioxidant properties, i.e., it cannot donate or receive electrons, but it interacts with numerous proteins that regulate the redox system [34]. Various proteins with an enzymatic function have been identified as requiring Zn to regulate different cellular processes [36,37]. Zn-containing metalloproteins are classified into three main groups: (1) metalloenzymes; (2) metallothioneins (MTs); and (3) gene-regulatory proteins [38]. Zn plays an important role in the redox system by modulating gene-regulatory proteins such as transcription factors and inducing the expression of MTs and antioxidant enzymes [39,40].

In the reproductive system, Zn performs essential functions in both the spermatogenic process and sperm maturation, as Zn deficiency in these organs has been reported to be associated with infertility [5,34,41]. Because Zn can interfere with Cd at different levels, including absorption, distribution, and excretion [31], Zn may exert favorable effects against Cd toxicity in the reproductive organs. However, it is not known whether Zn can benefit from the changes caused by Cd in the glycosylation of the plasma membrane of epididymal sperm. Therefore, this study aimed to analyze and determine whether Zn pretreatment can protect and enhance the glycosylation of the sperm membrane, endocrine activity, and antioxidant activity when exposed to Cd.

## 2. Results

### 2.1. Body and Reproductive Organ Weight

During the treatments, no clinical signs indicative of systemic toxicity were observed in either group. No significant differences were observed in the final body weights of the control group, (X ± SEM) 401 ± 5.3 g; the Cd-exposed group’s corresponding value was 399 ± 8.7 g, while the Zn-supplemented group’s was 399 ± 4.6 g, and the Zn + Cd group’s was 401 ± 5.8 g. No significant differences were observed in the weights of the testes and epididymis between the groups. In the control group, the weight of the testes was 1.9 ± 0.048 g, and the epididymis was 0.62 ± 0.027 g; in the Cd-exposed group, the values of the testes were 1.9 ± 0.056 g, and those of the epididymis were 0.61 ± 0.024 g; in the Zn-supplemented group, the testes weighed 1.9 ± 0.056 g, and the epididymis weighed 0.67 ± 0.0070 g, and for the group treated with Zn + Cd, the values for the testes were 2.0 ± 0.027 g, and those for the epididymis were 0.67 ± 0.014 g.

### 2.2. Concentration of Cd and Zn in Blood, Testes, and Epididymis

Table 1 shows the concentrations of Cd and Zn analyzed in blood, testes, and epididymis samples of control, Cd-exposed, Zn-supplemented, and Zn + Cd groups. In the blood, the Cd-exposed group showed a significant increase compared to the group and the treatments with Zn-supplemented and Zn + Cd groups (*p* < 0.05). In the testes, the group treated with Cd experienced a significant increase (*p* < 0.05) in the concentration of this metal compared to the control and those treated with Zn-supplemented and Zn + Cd groups. Similarly, in the epididymis, there was a significant increase (*p* < 0.05) in the concentration of Cd in the Cd-exposed group, with the region of the cauda showing the greatest accumulation when compared with the control and the Zn-supplemented and Zn + Cd groups. In the Zn + Cd group, Cd concentrations in the blood, testes, and epididymis were significantly lower (*p* < 0.05) than in the Cd-exposed group and significantly higher (*p* < 0.05) than in the control and Zn-supplemented groups. On the other hand, the concentration of Zn in the blood, testes, and caput epididymis in the Cd-exposed group decreased significantly (*p* < 0.05) compared to all the other groups. In the Zn-supplemented group, there were significant differences (*p* < 0.05) in the blood and both reproductive organs concerning the Cd-exposed group, while in the Zn + Cd group, there were no significant differences when compared with the control group; however, there was a significant increase (*p* < 0.05) in the testicle and epididymis among the Cd-exposed group.

### 2.3. Serum Testosterone Concentration

Table 2 shows the serum testosterone concentrations at 90 PD for the control and those that underwent Cd-exposed, Zn-supplemented, and Zn + Cd groups. The results showed a significant decrease (*p* < 0.05) in the testosterone concentration in the Cd--exposed group compared to the control group and the other groups. The Zn-supplemented and Zn + Cd groups did not show significant differences compared to the control group, but both groups showed a significant increase (*p* < 0.05) in the Cd-exposed group.

### 2.4. Histological Evaluation of Testes and Epididymis

Table 3 shows the morphometric, maturation index, and histopathological analysis of the seminiferous epithelium of the control and the Cd-exposed, Zn-supplemented, and Zn + Cd groups. The control group exhibited an undisturbed diameter, area, maturation index, and histopathologic index of seminiferous tubules (Table 3). The diameter and area, the maturation index, and the histopathological characteristics of the seminiferous tubules of the Cd-exposed, Zn-supplemented, and Zn + Cd groups did not show significant differences among themselves or when compared with the control group (Table 3). The histological sections of the control group (Figure 1A) showed the structure of an epithelium characterized by the presence of different types of developing germ cells, including spermatogonia (Sg), spermatocytes (Sp), round spermatids (Rs), elongated spermatids (Es), and Sertoli cells (SC), corresponding to complete spermatogenesis with a mature and healthy seminiferous epithelium. In the histological sections, the epithelium of the Cd-exposed (Figure 1B), Zn-supplemented (Figure 1C), and Zn + Cd (Figure 1D) groups showed the presence of all types of developing germ cells that constitute complete spermatogenesis of a mature and healthy seminiferous epithelium. In the interstitium of the Cd-exposed group (Figure 2B), damage to the testicular vascular endothelium was present, compared to the control (Figure 2A), Zn-supplemented (Figure 2C), and Zn + Cd (Figure 2D) groups, where no alterations were present.

Table 4 shows the morphometric analysis of the height and area of the epithelial cells of the epididymis in the three regions, and Figure 3 shows the histological sections of the epididymal regions of all groups. The height and area of the epididymal epithelium of the control group did not present alterations. In the histological sections of the different regions of the epididymis of the control group (caput (Figure 3A), corpus (Figure 3B), and cauda (Figure 3C)) was observed a pseudostratified epithelium characterized by the presence of different cell types, including principal cells (Pc), basal cells (Bc), and clear cells (Cc). Additionally, a significant number of sperm were present in the epididymal lumen. The Cd-exposed group showed a significant increase (*p* < 0.05) in the height and area of epididymal epithelium compared to all groups (Table 4). The histological sections of the epididymis (Figure 3D,E) showed an increase in epithelial height in the caput and corpus regions. The region of the cauda (Figure 3F) shows a hyperplastic epithelium of cribriform appearance, with increased vacuolization and dense vesicles in the cytoplasm. However, the presence of sperm in the lumen of the three epididymal regions was observed. In the Zn-supplemented group, the height and area of the epididymal epithelium remained unchanged (Table 4). In the histological sections of the different regions of the epididymis (Figure 3G–I), alterations were not present, and all the cell types are shown. In the Zn + Cd group, no alterations were observed in the height or area of the epithelium. In the cauda region, there was a significant increase in height and area compared to the control and Zn-supplemented groups. However, it was lower than in the Cd-exposed group (Table 4). In the caput and corpus (Figure 3J,K), there was also an epithelium without alterations, with the different cell types, slight vacuolization, and a high density of sperm in the epididymal lumen. However, in the cauda (Figure 3L), a slight hyperplasia was observed that increased the height of the epithelium and the presence of slight vacuolization compared to the control and Zn-supplemented groups. However, these changes were less pronounced compared to the Cd-exposed group, and the presence of sperm in this region was not affected.

### 2.5. Antioxidant Activity in the Testes and Epididymis

Figure 4A shows the antioxidant activity of SOD in the testes and epididymis. In the testes, the Cd-exposed, Zn-supplemented, and Zn + Cd groups showed an increase in SOD activity, but this was not significant when compared with all groups. In the epididymis, in the caput, the Cd-exposed group showed a significant decrease (*p* < 0.05) when compared with all groups. For the Zn-supplemented group, there were significant differences (*p* < 0.05) compared with the Cd-exposed group. The group treated with Zn + Cd exhibited a significant decrease (*p* < 0.05) when compared with the control group and a significant increase (*p* < 0.05) when compared with the Cd-exposed group. In the cauda region, the Cd-exposed group showed a significant decrease (*p* < 0.05) compared with all groups. The Zn-supplemented group showed a significant increase (*p* < 0.05) compared with the Cd-exposed group. The Zn + Cd-treated group showed a significant increase (*p* < 0.05) compared with the Cd-exposed group. Figure 4B shows the activity of CAT in both reproductive organs. The activity of CAT in the testes and the caput of the epididymis of the Cd-exposed group showed a significant decrease (*p* < 0.05) when compared with all groups. The Zn-supplemented and Zn + Cd groups did not show significant differences among themselves or with the control group, but they showed a significant increase (*p* < 0.05) compared to the Cd-exposed group. In the cauda, the Cd-exposed group showed a significant decrease (*p* < 0.05) compared with the control group. The Zn-supplemented group showed a significant increase (*p* < 0.05) compared with the Cd-exposed and Zn + Cdgroups. The group treated with Zn + Cd showed a significant decrease (*p* < 0.05) compared with the control group, but showed no significant differences with the Cd-exposed group. The GSH activity in the testes and epididymis is shown in Figure 4C. The Cd-exposed group showed a significant decrease (*p* < 0.05) in the testes and epididymis compared with all groups. The Zn-supplemented and Zn + Cd groups did not show significant differences among themselves or with the control group, but they showed a significant increase (*p* < 0.05) when compared with the Cd-exposed group in the testes and the caput region. In the cauda, the Zn + Cd group showed a significant decrease (*p* < 0.05) when compared with the Zn-supplemented group, but it was significantly higher (*p* < 0.05) when compared with the Cd-exposed group.

### 2.6. Sperm Parameters

Table 5 shows the sperm parameters of the three epididymal regions of the control and the Cd-exposed, Zn-supplemented, and Zn + Cdgroups. The Cd-exposed group showed a significant decrease (*p* < 0.05) in the three evaluated parameters compared to all groups in the three epididymal regions. The Zn-supplemented did not show significant differences compared to the control group. The Zn + Cd group showed significant differences (*p* < 0.05) in sperm concentration when compared with Cd-exposed and Zn-supplemented groups in the caput and corpus regions, in the cauda, a significant difference was observed when compared with all groups.

Regarding sperm viability, significant differences (*p* < 0.05) were found with the Cd-exposed group in the caput and corpus. In the cauda, significant differences (*p* < 0.05) were found among all groups. The morphological normality of the sperm showed significant differences (*p* < 0.05) in the caput and corpus compared with the Cd-exposed group; in the cauda region, it showed significant differences (*p* < 0.05) with all groups.

### 2.7. Carbohydrate Distribution and Fluorescence Index of N-Acetylglucosamine and/or Sialic Acid of the Sperm Membrane

The N-acetylglucosamine and/or sialic acid residues in the sperm membrane from the three epididymal regions of the control and the Cd-exposed, Zn-supplemented, and Zn + Cd groups were identified using WGA-FITC lectin. We identified four patterns of distribution of the carbohydrates in the sperm membrane (Figure 5). Pattern 1 shows the distribution of N-acetylglucosamine and/or sialic acid throughout the whole sperm; in Pattern 2, the distribution was mainly in the head region of the sperm, and in Pattern 3, it was in the head region, the middle part, and slightly in the main part, and in Pattern 4, there was no fluorescence.

Figure 6 shows the percentages of distribution patterns of N-acetylglucosamine and/or sialic acid of all groups. In the three epididymal regions of the control and treated groups, Pattern 2 was more dominant than Patterns 1, 3, and 4. However, in the Cd-exposed group, the sperm from the three epididymal regions, Patterns 1, 2, and 3, showed a significant decrease (*p* < 0.05) in the percentage of distribution compared to all treatments. Pattern 4 showed a significant increase (*p* < 0.05) in the percentage of sperm without fluorescence compared to all groups. The Zn-supplemented group did not show significant differences compared to the control group. The Zn + Cd group did not show significant differences for Patterns 1, 2, and 3 in the caput and corpus when compared to the control and Zn-supplemented groups; in the cauda region, there was a significant decrease (*p* < 0.05) in the distribution of Patterns 1, 2, and 3 when compared to the control and Zn-supplemented groups, but it was significantly higher (*p* < 0.05) than the Cd-exposed group. In addition, Pattern 4 also showed significant differences (*p* < 0.05) when compared with the control and Zn-supplemented groups, but it was inferior when compared with the Cd-exposed group.

Figure 7 shows the fluorescence index of N-acetylglucosamine and/or sialic acid in sperm from the three epididymal regions of the control, Cd-exposed, Zn-supplemented, and Zn + Cd groups. In the Cd-exposed group, there was a significant decrease (*p* < 0.05) in the fluorescence index in the three regions compared with all groups. The control group and the Zn-supplemented group did not show significant differences. In the Zn + Cd group, a significant increase (*p* < 0.05) was observed in the caput compared to all groups; in the corpus region, there were significant differences (*p* < 0.05) with the Cd-exposed group, and in the cauda, there was a significant decrease (*p* < 0.05) compared to the control and Zn-supplemented groups, but it was higher than that of the Cd-exposed group.

### 2.8. Carbohydrate Distribution and Fluorescence Index of Mannose in the Sperm Membrane

The mannose residues in the sperm membrane of the three epididymal regions of all groups were identified using the Con A-FITC lectin. Four distribution patterns were identified (Figure 8). Pattern 1 was present in the head and main part; in Pattern 2, it was present in the head and middle part, in Pattern 3, the distribution of mannose was mainly in the head, and in Pattern 4, there was no labeling.

Figure 9 shows the percentages of the four patterns of mannose distribution. Pattern 3, unlike patterns 1, 2, and 4, showed the greatest dominance in all epididymal regions of all groups. However, in the Cd-exposed group, the percentage distribution of mannose in the membrane of the sperm in the three epididymal regions showed a significant decrease when compared with all the groups. Pattern 4 showed an increase in the percentage of sperm without fluorescence. The Zn-supplemented group did not show significant differences compared with the control group. The Zn + Cd group did not show significant differences in Patterns 1, 2, and 3 in the caput and corpus compared with the control and Zn-supplemented groups. However, in the cauda region, there was a decrease in these patterns compared with those; it was significantly higher than in the Cd-exposed group. In addition, Pattern 4 also showed significant differences compared with the control and Zn-supplemented groups, but it was lower than that of the Cd-exposed group.

Figure 10 shows the mannose fluorescence index in the sperm membrane of the three epididymal regions of all groups. In the Cd-exposed group, there was a significant decrease (*p* < 0.05) in the mannose fluorescence index in the sperm membrane of the three epididymal regions compared with all groups. There were no significant differences between the control and the Zn-treated groups. In the Zn + Cd group, there were no significant differences in the mannose fluorescence index in the membrane of the sperm in the caput and in the corpus showed a significant increase (*p* < 0.05) compared with all groups; in the cauda, it showed a significant decrease (*p* < 0.05) compared with the control and Zn-supplemented groups, and it was higher in the Cd-exposed group (*p* < 0.05).

## 3. Discussion

Research on natural products, antioxidants, and essential trace elements with the potential to mitigate or prevent heavy metal-induced toxicity is an area of great interest. Recent studies have highlighted their role in reducing oxidative stress and protecting biological systems from heavy metal damage [42,43]. It is known that Zn is a potent antagonist of Cd toxicity; however, understanding the effects of Zn on the reproductive system and the interactions with Cd on the epididymal sperm maturation is necessary to comprehend the possible mechanisms through which Zn exerts its action against alterations in sperm glycosylation after exposure to Cd. In the present study, the determination of Zn and Cd in blood, testes, and epididymis showed differences in the concentration of these metals. The toxic effects of Cd depend on the doses, time, and method of administration, so the dose to which we exposed the animals from 35 to 56 PD was still present at 90 PD. This is explained in mammals by a limited capacity to respond to Cd exposure; i.e., it does not undergo metabolic degradation into less toxic molecules, and therefore, a low percentage is excreted [44]. Cd has a very long biological half-life, and exposure to it results in Cd entering the blood after absorption, binding to erythrocyte membranes, blood albumin, and MTs and thus being distributed throughout the body, where it accumulates in various target organs, causing adverse effects [44,45]. It has been shown that Cd can exert molecular mimicry by having a chemical structure resembling essential ions such as Zn, and it can displace it from its binding sites, which would explain the reduction in Zn levels in reproductive organs [46]. In animals previously treated with Zn, we observed that Cd concentrations decreased, confirming the importance of Zn in reproductive organs. Our results are similar to studies in which Zn can promote a reduction in Cd concentrations in the blood and reproductive organs [47,48], and it is known that the main mechanism of protection is through direct competition with Cd for absorption in the body and preventing it from binding to the active sites of proteins and enzymes [49]. Therefore, in our results, a possible mechanism of protection for Zn is due to the ionic radius of Zn (0.75 Å), which is similar to the ionic radius of Cd (0.95 Å). Thus, it allows Zn to interact, displace, and compete with Cd for different transport channels such as calcium and divalent metal transporters (DMT1) and thus prevent the cytotoxic effects of Cd [35,37]. However, it is still unclear whether the reduction in Cd accumulation achieved through Zn intake is significant for human health, and more evidence is needed to determine a causal relationship [50].

Cd is one of the most studied endocrine disruptors that alter the synthesis, secretion, and regulation of steroid hormones such as testosterone [51,52,53]. In our study, the analysis of serum testosterone concentration in adult rats (90 PD) exposed to Cd showed a decrease. Similar results have been reported in animals exposed to Cd [53,54]. In addition, Cd is known to reduce steroidogenic functions by negatively regulating the expression of several key enzymes and proteins for this process in Leydig cells [52], including steroidogenic acute regulatory protein (StAR) RNA, the enzymes 3β- and 17β-hydroxysteroid dehydrogenase [55], cytochrome P450 (P450SCC), cytochrome P45017A1 (P450c17), and androgen receptors [56]. It is also known that Cd induces apoptosis and alters the development and number of Leydig cells, which is associated with a decrease in testosterone synthesis and secretion [57]. Also, it has been found that Cd inhibits the secretion of gonadotropins from the adenohypophysis, such as LH and FSH. In this way, Cd can induce changes in the steroidogenic and spermatogenic activity of the testes [58,59]. In the group treated with Zn + Cd, we observed a maintained level of testosterone concentrations; similar studies point out that the administration of Zn in animals exposed to Cd restores the testosterone concentration and that Zn deficiency is one of the causes of low levels of this androgen, in addition to inducing damage to the reproductive organs [47,60]. It has been reported that the Zn supply in Leydig cells is required for steroidogenic regulation through the expression of Zn transporters, such as ZnT-8. In addition, Zn plays an important role in the phosphorylation of StAR to carry out the modulation of this process [61].

It has been shown that Cd at high doses (2 mg/kg/bw) can cause testicular atrophy, a reduction in the diameter and area of the seminiferous tubules, the arrest of spermatogenesis, the loss and shedding of germ cells, increased vacuolation, edema, and hemorrhage, as reported in previous studies [58,62,63]. However, this was not the case for us, as we did not find any changes in testicular weight or morphometric analyses of seminiferous tubules when administering a lower dose (0.5 mg/kg/bw). However, changes were observed in the vascular endothelium in the interstitium. It has been shown that the vascular endothelium is one of the main target tissues for Cd toxicity, even at low doses [64,65,66]. Cd damages the internal spermatic artery and the pampiniform plexus in the testis [40,66,67], which causes an increase in vascular permeability, resulting in fluid leakage into the interstitium, edema, hemorrhage, hypoxia, inflammation, and testicular necrosis [68,69]. In addition, Cd causes damage to the endothelial cells, where it has a direct effect on the adherent junctions between blood vessel cells, inducing alterations in the expression and function of the vascular endothelial cadherin, and it could cause a reorganization of the actin cytoskeleton [46,67], which could explain what we observed in our results. As for the analysis of the group treated with Zn + Cd, we observed beneficial effects on the vascular endothelium of the testis since there was no damage to the interstitium, like occurrences in Leydig cells, as reported previously [61].

The epididymis is an organ whose effects of exposure to heavy metals such as Cd have been little studied. Consequently, the magnitude of its effects on epididymal physiology is not fully understood [70]. In our analysis of the epididymal epithelium, we observed that Cd caused cellular hypertrophy in the caput and corpus regions, as well as cribriform hyperplasia in the cauda. A decrease in the epididymal lumen, an increase in the height and area of the epithelial cells, and an increase in vacuolization were also observed; however, the presence of sperm was evident in all three regions. Our results are consistent with studies that have determined the toxic effects of Cd on the structure and function of the epididymis [29,71,72]. Cd can cross the blood–epididymal barrier (BEB) and induce changes in its cellular structure and function. In addition, Cd can cause an increase in vacuolization, which can induce cellular disorganization of the epithelium [72]. The mechanism through which Cd induces the vacuolization of the epididymal epithelium is unclear, but it is known to be associated with several degenerative changes that include the accumulation of fluids, phospholipids, lipids, and glycoproteins. Vacuolization is characterized by the presence of large or small transparent vacuoles within and between epithelial cells, which can alter epithelial organization and displace the nucleus to the periphery of the cell [73]. The cribriform appearance with hyperplasic changes in the epithelium causes it to fold in on itself, forming pseudoglandular structures. These changes are associated with Cd toxicity, and the folding may, in turn, be a secondary response to the decrease in sperm and luminal fluid causing an altered microenvironment [73]. The epididymis is an organ that depends on androgens to maintain its histoarchitecture, so androgen deprivation may be associated with alterations in its development, morphology, and function [29]. Therefore, in our study, the low concentration of testosterone observed in the Cd-treated animals could explain the structural changes generated in the epididymal epithelium, although there is no definitive cause that can accurately explain the cribriform changes of the epididymis [73]. On the other hand, in this study, in animals treated with Zn + Cd, we observed beneficial effects due to the pretreatment with Zn supplementation, which consisted of protecting the epididymal epithelium in the caput and corpus regions. Thus, there was no structural damage, cellular disorganization of the epithelium, or increased vacuolization, in addition to the presence of sperm in the epididymal lumen, confirming the protective effect of Zn. The epididymis is an organ rich in Zn concentrations; therefore, its function in the maintenance of its structure and the secretory function of the epithelial cells is reflected by the presence of this trace element [74]. As for the region of the epididymal tail, we noticed that the hyperplasia decreased. The damage in this area could be due to the effort of the antioxidant response, which does not seem to be entirely sufficient to repair the damage that can occur during sperm storage in the face of Cd-induced alterations, and the secretory activity of the epididymal epithelium may have been forced to metabolic changes in the luminal compartment, which could accelerate the attempt to restore the parameters and ensure the survival of the sperm [72].

It has been shown that one of the main mechanisms involved in Cd toxicity is the induction of oxidative stress [75]. Cd alone cannot generate free radicals, but its toxicity is mediated through indirect mechanisms by interfering with the binding sites of metals such as Zn, selenium (Se), and manganese (Mn), among others, which are cofactors in antioxidant enzymes, thus altering the redox function [76], and enzymes such as SOD [40], CAT, GSH, glutathione reductase (GSR) and glutathione peroxidase (GPx) [52,77,78]. Cd can displace copper (Cu) and iron (Fe) from membrane proteins, which induce the Fenton reaction and promote lipoperoxidation [75,79]. In the present study, Cd caused alterations in the antioxidant activities of SOD, CAT, and GSH in the testes and epididymis. In the testes, SOD activity increased, unlike in epididymis where it decreased, while CAT and GSH activity decreased in both organs. Similar studies reported results like ours [60,80,81]. This increases ROS produced via Cd induces oxidative stress, causing an imbalance in the antioxidant defense system [82]. The activity of SOD in the testis could be the first line of defense compromised against Cd, which contains the different isoforms of SOD, such as cytosolic (Cu/Zn), mitochondrial (Fe/Mn), and extracellular (SOD-Ex), produced by both Sertoli cells and germ cells [83,84]. Regarding CAT and GSH in the testis and epididymis in our study, a decrease in their activity was observed in the testis because CAT has limited activity [83], while in the case of the epididymis, it has been shown that CAT activity is higher [62]. The low activity of SOD and CAT in the epididymis could be explained by the interference of Cd in the catalytic sites of cofactors in antioxidant enzymes [46]; in addition to that, the epididymis has another variety of antioxidant defenses, such as GPX5, thioredoxins, and sulfiredoxins, which have higher redox activity [7,85]. The decreased GSH activity in the testis and epididymis in our study can be explained by the interactions that Cd has with other molecules, by the affinity to bind to thiol groups and form complexes such as Cd-GSH [75,86]. In the Zn + Cd group, we observed an increase in SOD, CAT, and GSH activity in both organs, which is consistent with similar studies on the beneficial effects of Zn supplementation via increases the activity of the antioxidant system [34,87]. This can be explained by different mechanisms of Zn, since Zn ligands can interact with many proteins and enzymes to perform a variety of important functions, including modulation in the redox system [34]. The binding of Zn on proteins is through amino acids, cysteines (Cys), histidines (His), aspartic acid (Asp), and glutamic acid (Glu), which have been identified as the primary binding partners for Zn. The Cys-Zn interaction occurs at the structural Zn sites, while the interaction with Glu and Asp acids occurs at the catalytic Zn sites. Finally, the His–Zn interaction occurs at both Zn binding sites [37,39].

In the cytosolic metalloenzyme SOD (Cu/Zn), Cu is responsible for the catalytic activity, while Zn plays the role of regulator of the catalytic activity and stabilizer of its structure. The positive charge of the Zn attracts the superoxide radical, dismutating it into hydrogen peroxide [88]. One of the important mechanisms of Zn is its ability to bind strongly to MTs, thus playing an important role in balancing excessive amounts of metals and providing cellular protection, as well as representing one of the main detoxification mechanisms for Cd and preventing it from interfering with the redox system [34,89]

Thionein-MTs play an important role in Zn homeostasis, mainly in the distribution and storage. The binding of seven Zn ions on their Cys residues gives rise to MTs and regulates their function by scavenging free radicals and detoxifying heavy metals [39]. Zn modulates the metal regulatory transcription factor (MTF-1), a protein containing Zn finger domains. Zn binds to MTF-1, this complex translocates to the nucleus and binds to the metal response elements (MREs), thereby increasing the transcription of thionine-MTs and some Zn transporters [39,40]. Zn modulates another transcription factor, nuclear erythroid factor 2 (Nrf2); after the junction, both are translocated to the nucleus and finally bind to antioxidant response elements (AREs), thus activating the upregulation and transcription of several antioxidant proteins, such as the expression of peroxiredoxins, CAT [90], in addition, of glutamate–cysteine ligase, which modulates the rate of GSH synthesis and protects cells from free radicals, lipid hydroperoxides, and heavy metals [91].

In the present study, the sperm parameters (concentration, viability, and morphological normality) of the three epididymal regions in the group treated with Cd showed changes that are consistent with other studies [29,71]. Several mechanisms can explain the reduction in sperm quality induced via Cd [92], including its direct toxic effects on the testis, leading to reduced sperm concentration, viability, and morphological normality. High concentrations of Cd can cause atrophy of seminiferous tubules, a severe detachment of immature germ cells, and the detachment of immature germ cells, resulting in impaired spermatogenesis [92]. The decrease in the quality of sperm parameters correlates with the effects on testosterone concentration and oxidative stress, as we observed in our results, where Cd caused a decrease in testosterone; this would explain the low number of sperm entering the epididymis [71]. Furthermore, Cd has been reported to disrupt BEB integrity and alter the structure and function of the epididymal epithelium, leading to changes in the composition of the luminal fluid and the sperm themselves, which may affect sperm quality and maturation [73]. Sperm have a high content of polyunsaturated fatty acids in their membrane, which makes them susceptible to Cd-induced oxidative damage that could affect sperm-quality parameters [93]. In the group treated with Zn + Cd, sperm concentration, viability, and morphology normality were unaffected in the caput and corpus regions; however, in the cauda region, the parameters decreased compared to the control group but were higher than in the group treated with Cd. These findings are consistent with previous studies showing that Zn supplementation decreased the damage caused by Cd exposure when administered together [31,48,81]. In our study, Zn supplementation was administered before Cd exposure, likely conferring a protective effect by maintaining epithelial integrity and thus favoring sperm quality parameters in the caput and corpus region, but not in the cauda region, where the parameters decreased, possibly due to the histological damage that persisted in this region, exposing the sperm to a modification of the luminal microenvironment during storage and Cd-induced oxidative stress [72].

Until now, the role of Zn supplementation in the glycosylation process during epididymal sperm maturation in animals exposed to Cd was unknown. In the present study, we report the effects of Zn on the distribution and abundance of carbohydrates such as N-acetylglucosamine and/or sialic acid and mannose in the sperm membrane during its journey through the epididymis. N-acetylglucosamine is required for the acquisition of motility [94], and sialic acid confers a negative charge on the sperm membrane, suggesting that it is involved in the interaction of the spermatozoon with the zona pellucida [95]. In our results, we observed that the distribution and presence of N-acetylglucosamine and/or sialic acid in sperm obtained from the three regions of the epididymis were observed in the region of the head and intermediate zone of the sperm, where they were most abundant, as reported in previous studies [14,96]. Similarly, the fluorescence index of N-acetylglucosamine/sialic acid was higher in the corpus region of the epididymis. In the sperm of the group treated with Cd, the distribution of these glycans was altered, as was the fluorescence index in the three regions of the epididymis, as reported in the studies of [29]. In the epididymis, it has been studied that the synthesis and secretion of glycosyl hydrolase and glycosyltransferase enzymes are responsible for the cleavage and addition of terminal sugar residues from the glycoconjugates of the sperm membrane [13]. It has been reported that N-acetylglucosamine residues are added by the enzyme N-acetylglucosaminyltransferase from the distal part of the caput to the proximal region of the epididymis cauda, while sialic acid is added via sialyltransferase, which is more active in the proximal region of the caput and less active in the caudal region [11,13], suggesting that it is the body of the epididymis where the highest activity of these enzymes is found, which is consistent with what we observed in our results. Mannose also plays a very important role in the glycoproteins of the sperm membrane since its presence allows the development of the process of sperm capacitation [97], the interaction of the spermatozoon with the zona pellucida, and the acrosomal reaction [98]. In our study, mannose was distributed in a higher percentage in the region of the head and the intermediate zone of the sperm, with a higher fluorescence index in the caput and corpus regions of the epididymis, which is consistent with previous studies [29,99,100]. The changes observed in the distribution and presence of N-acetylglucosamine and/or sialic acid and mannose in the plasma membrane of sperm in our study could be explained by the decrease in testosterone concentration caused by Cd. The possible mechanism of toxicity of Cd in the epididymis, which, being an organ dependent on androgens for its development and function, may be affected by exposure to Cd, which alters the synthesis and secretion of testosterone so that the inhibition or low concentration of testosterone reaching the epididymis affects the synthesis of glycosylating enzymes and, consequently, glycosylation [29]. In our study, in the group with Zn + Cd, we observed that Zn supplementation favored sperm glycosylation by maintaining and protecting the distribution and presence of N-acetylglucosmine and/or sialic acid and mannose in the caput and corpus. However, in the cauda, where we had described a slight alteration of the epithelium, the distribution and presence of these carbohydrates showed a slight decrease, but there was a significant recovery compared to the Cd-exposed group. Therefore, Zn plays an important role in the glycosylation of the sperm membrane during its journey through the epididymis, and this is related to what we observed in our study, where Zn supplementation favored testosterone synthesis in the testis and, in turn, favorably influenced the glycosylation function that takes place in the epididymis. Another explanation for the beneficial effects of Zn supplementation in our results is that it is the only metal that is present in all types of enzymes for its activity, including glycosidases and glycosyltransferases, which are fundamental in the glycosylation process [101]. Also, Zn competes with Cd for entry into the epididymal epithelium, preventing it from interacting with certain proteins and enzymes necessary for sperm maturation, and it even promotes the expression of MTs [75] and increases antioxidant activity, as we observed in our results, since Zn inhibits the toxic effects of Cd in the epididymis, thus favoring reproductive fertility.

## 4. Materials and Methods

### 4.1. Animals

Pregnant female Wistar rats were obtained from the Animal Facility of the Universidad Autónoma Metropolitana-Iztapalapa. Only newborn males were used in the experiments, and the day of birth was considered the first day of postnatal life (PD). On day 21, all offspring were weaned, maintained on a 12 h light–dark cycle (lights on at 8:00 p.m. and off at 8:00 a.m.), and given food (rat chow, Harlan Laboratories, Indianapolis, IN, USA) and water ad libitum. All experiments were carried out in accordance with the standard guidelines outlined in the Guide for the Care and Use of Laboratory Animals (NIH, 2011) and Mexico’s Official Norm (NOM-062-ZOO-1999, reviewed in 2001).

### 4.2. Treatments

Newborn male rats were randomly divided and housed in four groups with *n* = 7 pups per dam. The first group was used as a control and received 100 μL of vehicle (saline solution), the second group received 0.5 mg/kg/100 µL of body weight (bw) of Cd (CdCl_2_, Sigma-Aldrich, St. Louis, MO, USA) diluted in saline solution, the third group received 1 mg/kg/100 µL of bw of Zn (ZnCl_2_, Sigma Aldrich, City of Mexico, Mexico), diluted in saline solution, and the fourth group received Zn + Cd at the respective doses. The administration of Zn supplementation started from 1–56 PD and the administration of Cd exposure from day 35–56; the same administration schedule was followed for the Zn + Cd group. The doses were administered intraperitoneally. Doses and administration were chosen based on previous studies [29,30,31,54,80]. The administration of Zn from the day of birth is suggested as a preventive strategy to assess its potential protective effect against Cd exposure, which included days 35 and 56 of life. This period is crucial for reproductive development, as it aligns with the increase in testosterone levels, the initiation of the first wave of spermatogenesis, and the maturation of the epididymal epithelium. On day 90, the animals were euthanized to collect blood, the testes, and the epididymis for subsequent analysis.

### 4.3. Experimental Procedure

At 90 days of age, animals were weighed and euthanized via decapitation under anesthesia. Peripheral blood samples were collected in two metal-free Vacutainer tubes. Cd and Zn concentrations in the blood were analyzed through atomic absorption spectrophotometry from tubes containing EDTA-K2. Blood serum was collected via centrifugation to determine testosterone levels through an enzyme-linked immunosorbent assay (ELISA). Each sample was quantified in duplicate using a commercially available kit (Testosterone ELISA EIA-1559, DRG, Springfield, NJ, USA). The epididymides and testes were dissected bilaterally and weighed; 100 mg of each tissue was used to quantify Cd and Zn via atomic absorption spectrophotometry, histological processing was performed, and the antioxidant activity of SOD, CAT, and GSH was determined, while sperm were collected from the epididymis to assess concentration, viability, and morphology. In addition, lectin markers were used to evaluate the distribution patterns of N-acetylglucosamine and/or sialic acid and mannose via a fluorescence index using lectins labeled with fluorescein isothiocyanate (FITC); see Figure 11.

### 4.4. Biochemical Analysis

#### 4.4.1. Quantification of Cd and Zn in Blood, Testes, and Epididymis

Approximately 3 mL of peripheral blood samples were stored in Vacutainer tubes at 4 °C, and testicular and epididymal tissue samples were stored in polypropylene tubes at −20 °C; 200 µL of each blood sample was collected, and 800 µL of a mixture of nitric, perchloric, and sulfuric acids (5:2:1) was added. Additionally, 500 µL of concentrated nitric acid (Suprapure, Merck, City of Mexico, Mexico) was added to the tissue samples. After 7 days in the acid solution, the blood and tissue samples were analyzed in a Perkin-Elmer AA 600 atomic absorption spectrophotometer and a Cd and Zn P-E hollow cathode lamp from an AS 800 graphite furnace set at a wavelength of 228.8 nm. For each analysis, calibration curves were constructed using aqueous Cd and Zn reference standards (0.5, 1.0, 2.0, 4.0, and 6.0 µg/L) (GFAA Stock mixed standard, Perkin Elmer, City of Mexico, Mexico). To optimize and validate the measurements, Cd and Zn quality-control measurements were performed at the beginning and end of each analysis using a previously digested Standard Reference Material 1577b, Bovine Liver, National Institute of Standards Technology (NIST 1577b, USA) [102]. Concentrations of Cd and Zn were expressed as µg/mL in blood samples and as µg/g in tissue samples.

#### 4.4.2. Quantification of Serum Testosterone

Blood was collected from the bodies of the individuals in tubes (BD Vacutainer SST, Mexico City, Mexico) with separation gel; it was centrifuged for 15 min at 3000 rpm, and the serum was collected. Free testosterone was quantified through an enzyme-linked immunosorbent assay (ELISA). Each sample was quantified in duplicate using a commercially available kit (Testosterone ELISA EIA-1559, DRG). Each microtiter well was coated with an antibody directed against an antigenic site on the testosterone molecule, such that endogenous testosterone in the samples competed with a horseradish peroxidase–testosterone conjugate (T-HRP) to bind to the coated antibody. After incubation, the unbound conjugate was removed through washing. The amount of bound peroxidase conjugate was inversely proportional to the concentration of free testosterone in the sample. Samples were run in duplicate and read using a spectrophotometer at an optical density of 450 nm. After the addition of the substrate solution, the color intensity obtained was inversely proportional to the concentration of testosterone in the sample and was quantified using a standard curve.

#### 4.4.3. Processing and Histological Evaluation of Testes and Epididymis

After the extraction of the testes and regionalized epididymis, they were fixed in modified Karnovsky’s solution for 24 h and processed for embedding in epoxy resin (EPON 812 Ted Pella, Inc., Redding, CA, USA). This was followed by four 30 min washes with a 0.1 M sodium cacodylate buffer. This was followed by post-fixation with 1% osmium tetroxide (Sigma Aldrich, St Louis, MO, USA) for one hour, followed by three washes with distilled water. The samples were dehydrated in a series of alcohols: 60%, 70%, 80%, 90%, 96%, up to absolute alcohol (two 10–15 min washes for each alcohol). To begin the resin embedding process, two 20 min washes with propylene oxide were performed. Resin impregnation was then performed, with the specimens left in gradual dilutions of propylene oxide: EPON (2:1 for 1 h, 1:1 for 24 h, 1:2 for 24 h) and pure EPON for 24 h. Each tissue section was placed in molds with pure EPON for 24 h in an oven at 60 °C for polymerization. Semi-thin sections of 1 µm thickness were cut with an ultramicrotome (Leica model Ultracut-UCT, Leica, Zurich, Switzerland) and stained with 0.5% toluidine blue (Sigma Aldrich, Mexico). Histological analysis was performed using a Leica microscope at 20× and 60× with an image analysis system (Image-ProPlus 6.0), morphometric analysis (diameter and area) of the seminiferous tubules, a maturation index [103], and a histopathological index [104].

#### 4.4.4. Determination of Antioxidant Activity in the Testes and Epididymis

The antioxidant activities of SOD, CAT, and GSH were determined using specific kits (Cayman Chemical, City of Mexico, Mexico), and testicular and epididymal tissue samples were homogenized according to each assay. Cayman’s SOD assay kit uses a tetrazolium salt to detect superoxide radicals generated via xanthine oxidase and hypoxanthine. One unit of SOD is defined as the amount of enzyme required to exhibit 50% dismutation of the superoxide radical. Cayman’s CAT Assay Kit utilizes the peroxidative function of CAT to determine enzyme activity. The method is based on the reaction of the enzyme with methanol in the presence of an optimal concentration of H_2_O_2_. The resulting formaldehyde is measured colorimetrically using 4-amino-3-hydrazino-5-mercapto-1,2,4-triazole (Purpald) as a chromogen.1,2 Purpald specifically forms a bicyclic heterocycle with aldehydes that turn from colorless to purple upon oxidation. Cayman’s GSH Assay utilizes a carefully optimized enzymatic recycling method using glutathione reductase to quantify GSH. The sulfhydryl group of GSH reacts with DTNB (5,5′-dithio-bis-2-(nitrobenzoic acid)) to form a yellow-colored 5-thio-2-nitrobenzoic acid (TNB). The resulting mixed disulfide, GSTNB (between GSH and TNB), is reduced through glutathione reductase to recycle the GSH and produce more TNB. The rate of TNB production is directly proportional to this recycling reaction, which, in turn, is directly proportional to the concentration of GSH in the sample. Measuring the absorbance of TNB at 405–414 nm provides an accurate estimate of GSH in the sample.

#### 4.4.5. Collecting Sperm Samples from the Epididymis and Sperm Determination

Each region of the epididymis was placed in an Eppendorf tube containing 500 μL of Ringer’s solution at 36 °C; sperm extraction was performed using the cut and filter technique according to [29,105]. The final suspension obtained from washing the sperm from each region of the epididymis was used for sperm determination, with the values reported being expressed concerning the total sperm present in 1 mL of Ringer’s solution.

##### Sperm Concentration

A 1:50 *v*/*v* dilution was performed in Eppendorf tubes of the sperm obtained and washed with distilled water (980 μL of distilled water and 20 μL of the sperm sample were applied). For counting, a Neubauer chamber was used where an aliquot of 10 μL of the sperm–water dilution (1:50 dilution) was placed on each side and allowed to stand for 1 min. Counting was performed on both sides of the chamber using an Olympus BX 51 optical microscope (Tokyo, Japan).

The number of sperm present in the first upper left frame of the chamber was counted. According to the number of sperm counted in the first upper square were the squares to be counted from the Neubauer chamber. Five frames were counted from each side of the chamber. The sperm concentration is expressed in millions/mL, for which the mean of the count of both sides of the chamber was obtained and divided by the corresponding conversion factor.

##### Sperm Viability

Five μL of eosin–nigrosin solution was placed on a slide, as well as five μL of the sperm sample. They were mixed and then smeared along the slide and allowed to dry at a temperature of 36.5 °C on a thermoplate. They were observed under an Olympus B× 51 optical microscope (Tokyo, Japan), at 40×, and the number of stained (dead) or unstained (live) cells was counted with the help of a laboratory counter. One hundred sperm were counted, and the number of live sperm was represented as a percentage of vitality.

##### Sperm Morphology

A 10 μL aliquot of the sperm sample was taken, placed on a slide, and smeared along the slide. It was left to dry for 5 min at room temperature, and the samples were then stained using a rapid staining kit (Eperma-form ESF-076, FertiMexico, City of Mexico, Mexico). They were placed in fixative for 3 min and then placed in dye A for 2 min and dye B for 4 min, and they were rinsed with distilled water and allowed to dry. They were observed under an Olympus BX 51 optical microscope (Tokyo, Japan) at 40×. They were evaluated under the following criteria: The sperm should present a hook-shaped head; cases of presenting acinuate, pyriform, vacuolated, or double heads, or combinations among these, were considered abnormalities. The intermediate piece presented thinness and represented less than one-third of the width of the head and aligned with the longitudinal axis of the head. The flagellum was free of cytoplasmic droplets, folds, breaks, forks, or coils. The concentration (10^6^/mL), the percentage of viability, and the morphological normality were evaluated according to the criteria established by the WHO [106].

#### 4.4.6. Determination of Carbohydrate Distribution and Fluorescence Index in Sperm Membrane

Approximately 3 × 10^6^ sperm cells from each region of the epididymis were incubated with fluorescein isothiocyanate (FITC)-labeled lectins (Sigma Aldrich, St. Louis, MO, USA) to detect specific carbohydrates on the sperm membrane. The lectins used in this study were *Triticum vulgaris* agglutinin (WGA), which recognizes N-acetylglucosamine and/or sialic acid, and *Canavalia ensiformis* agglutinin (Concanavalin A, Con-A), which recognizes D-mannose.

Each sperm sample was incubated with 5 µL of the lectins, previously diluted 1:50 in PBS, for 30 min at 36 °C. The samples were then washed with Ringer’s solution via centrifugation at 500× *g* for 5 min, fixed in 1% paraformaldehyde for 1 h at room temperature, and finally washed with PBS before analysis.

#### 4.4.7. Evaluation of Carbohydrate Distribution via Epifluorescence Microscopy in Membrane Sperm Cells

The lectin-stained sperm samples were examined on an epifluorescence microscope (Olympus B×51) using a specific filter for FITC (460 nm excitation, 490 nm fluorescence, and 520 nm barrier filters). After different carbohydrate residue patterns were identified via lectin labeling on the sperm membrane, 100 sperms were counted and classified according to the patterns identified. A fluorescence control was included in each sample.

#### 4.4.8. Determination of Fluorescence Index via Flow Cytometry in Sperm Cells

Sperm samples containing 10^6^ cells, previously incubated with lectins, were analyzed in a FACScan flow cytometer (Becton Dickinson, Immunocytometry System, San Jose, CA, USA). Ten thousand events were stored and analyzed. For each sperm sample, the fluorescence index was obtained from histograms of fluorescence intensity versus the number of labeled cells. For each sample, a control sample was prepared by pre-incubating the lectins with the corresponding 0.3 M carbohydrate inhibitor for 30 min. Sperm were then added to this solution, and 10,000 events were analyzed by flow cytometry.

#### 4.4.9. Statistical Analysis

All data are presented as means ± standard errors of the mean (SEM). The normality of the data distribution was assessed using the Shapiro–Wilk test. Subsequently, comparisons among the four were analyzed through a one-way ANOVA, followed by Tukey’s post hoc test. The differences considered were the control vs. Cd-exposed vs. Zn-supplemented vs. Zn + Cd and the comparison between the groups, Zn-supplemented vs. Cd-exposed, Zn-supplemented vs. Zn + Cd, and Cd-exposed vs. Zn + Cd, for each parameter analyzed. *p* < 0.05 was considered statistically significant. All analyses were performed using GraphPad Prism version 8.01.

## 5. Conclusions

Zn is an essential micronutrient for testicular and epididymal function. It plays a critical role in spermatogenesis, testosterone production, maturation, storage, and the structural integrity of the sperm, in addition to modulating the redox system. Therefore, Zn supplementation has a strong capacity to prevent or mitigate the damage caused by Cd exposure to the reproductive system. Taken together, these findings highlight the importance of Zn supplementation in glycosylation and justify its consideration in nutritional and therapeutic strategies to optimize testicular and epididymal function.

## Figures and Tables

**Figure 1 ijms-26-04589-f001:**
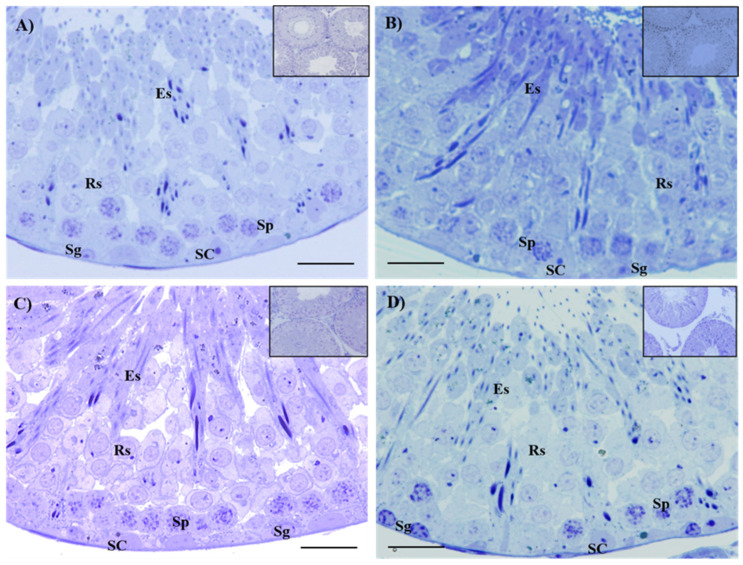
Histological sections of seminiferous tubules in the testes show the presence of cell types involved in spermatogenesis. The control is shown in (**A**), while the Cd-exposed is shown in (**B**), the Zn-supplemented in (**C**), and the Zn + Cd in (**D**) groups. The histological images contained the following cell types: spermatogonia (Sg), spermatocytes (Sp), round spermatids (Rs), elongated spermatids (Es), and Sertoli cells (SC); toluidine blue, 60×. Bar 30 µm.

**Figure 2 ijms-26-04589-f002:**
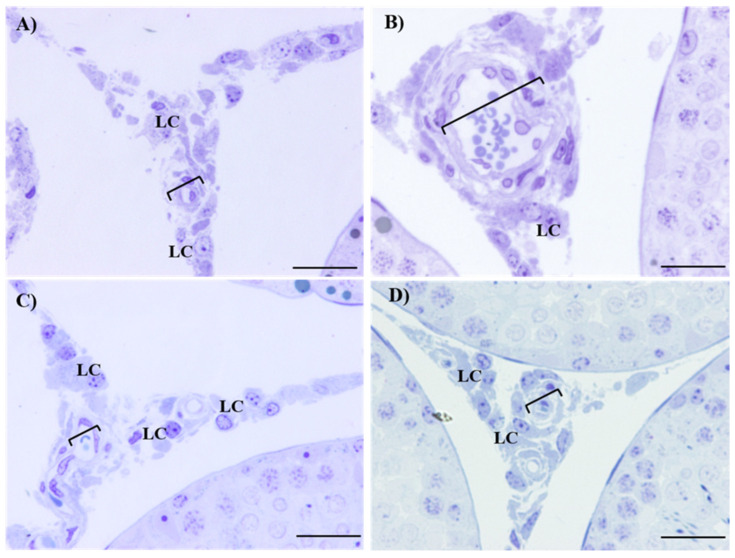
Histological sections of the testes interstitium. Leydig cells and some blood vessels with endothelial damage due to Cd treatment are observed as shown in (**B**) in comparison with the control (**A**), the Zn-supplemented (**C**), and the Zn + Cd (**D**) groups. Leydig cell (LC); bracketed bar indicates (blood vessels); toluidine blue, 60×. Bar 30 µm.

**Figure 3 ijms-26-04589-f003:**
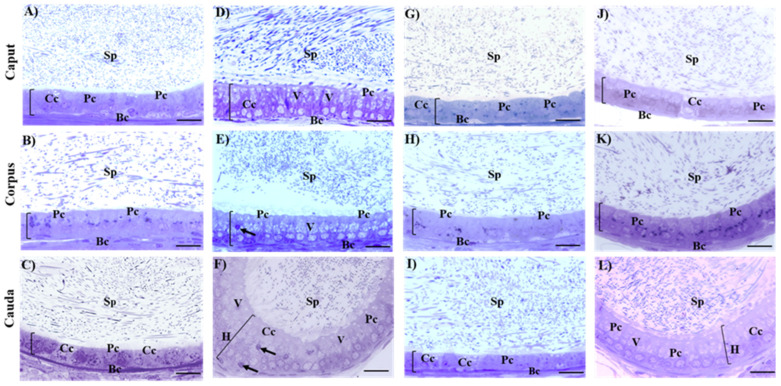
Histological sections of the epididymal regions. The characteristic cell types of the epididymal epithelium are seen, providing a luminal environment for the protection and maturation of sperm. Cd exposure caused an increase in epithelial height, vacuolization, and hyperplasia. In the Zn + Cd group, Zn protected the caput and corpus regions and reduced damage to the cauda. Control: (**A**) caput, (**B**) corpus, (**C**) cauda; Cd-exposured: (**D**) caput, (**E**) corpus, (**F**) cauda; Zn-supplemented: (**G**) caput, (**H**) corpus, (**I**) cauda; and Zn + Cd: (**J**) caput, (**K**) corpus, (**L**) cauda groups. Principal cell (Pc); basal cell (Bc); clear cell (Cc); sperm (Sp); hyperplasia (H); epithelial height (bar in brackets); vacuolization (V); dense vesicles (arrows); toluidine blue, 60×. Bar 20 µm.

**Figure 4 ijms-26-04589-f004:**
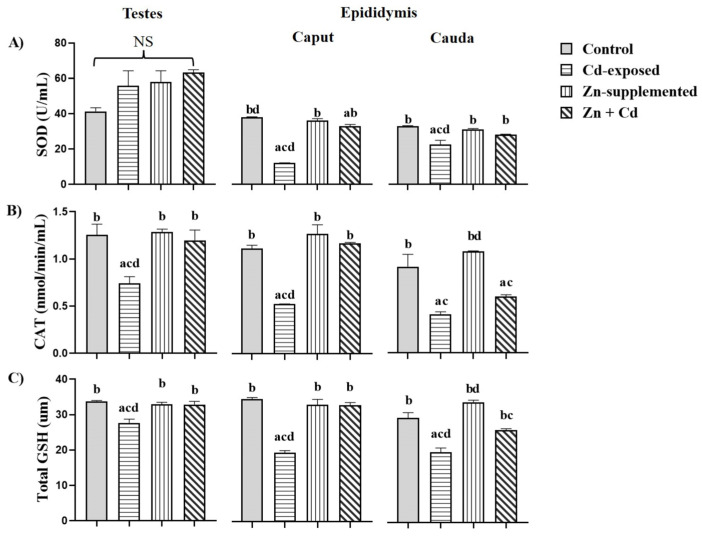
Antioxidant activity of the testes and epididymis. The activity of SOD, CAT, and GSH was altered in both the testes and the epididymis with Cd-exposure. However, in the Zn + Cd, Zn-supplementation favored the antioxidant activity in both organs. SOD (**A**), CAT (**B**), and GSH (**C**). Data expressed as means ± SEM for all analyzed groups (*n* = 7/treatment group). ^a^ *p* < 0.05 indicates differences vs. the control group; ^b^ *p* < 0.05 indicates differences vs. the Cd-exposed group; ^c^ *p* < 0.05 indicates differences vs. the Zn-supplemented group; ^d^ *p* < 0.05 indicates differences vs. the Zn + Cd group. No significant difference (NS).

**Figure 5 ijms-26-04589-f005:**
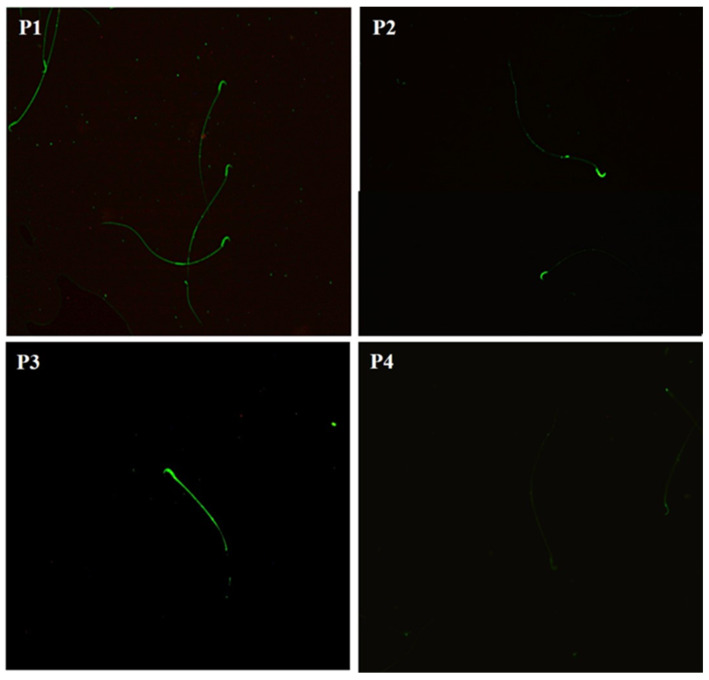
The photomicrograph shows the distribution patterns of fluorescence of N-acetylglucosamine and/or sialic acid in sperm. The sperm showed different patterns of fluorescence along their plasma membranes. Pattern 1 (P1): fluorescence throughout the spermatozoon; Pattern 2 (P2): fluorescence in the head; Pattern 3 (P3): fluorescence in the head and midbody; Pattern 4 (P4): sperm without fluorescence. Fluorescence microscopy, 20×.

**Figure 6 ijms-26-04589-f006:**
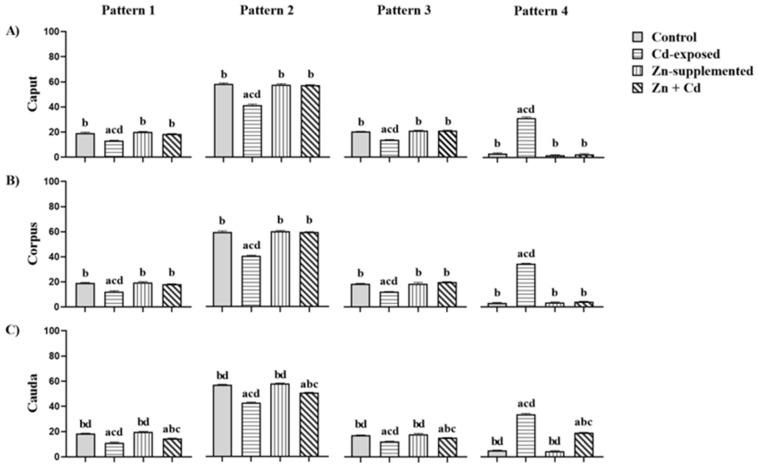
Percentages of distribution of fluorescence patterns of N-acetylglucosamine and/or sialic acid in the membrane of sperm in the three epididymal regions. The fluorescence distribution of Pattern 2 predominated in all regions, among treatments, the Cd-exposed group showed a decrease in the distribution of Patterns 1, 2, and 3 and an increase in Pattern 4. In the Zn + Cd group, Zn supplementation favored the distribution of N-acetylglucosamine and/or sialic acid. Data are expressed as means ± SEM for all analyzed groups (*n* = 7/treatment group). ^a^ *p* < 0.05 indicates differences vs. the control group; ^b^ *p* < 0.05 indicates differences vs. the Cd-exposed group; ^c^ *p* < 0.05 indicates differences vs. the Zn-supplemented group; ^d^ *p* < 0.05 indicates differences vs. the Zn + Cd group.

**Figure 7 ijms-26-04589-f007:**
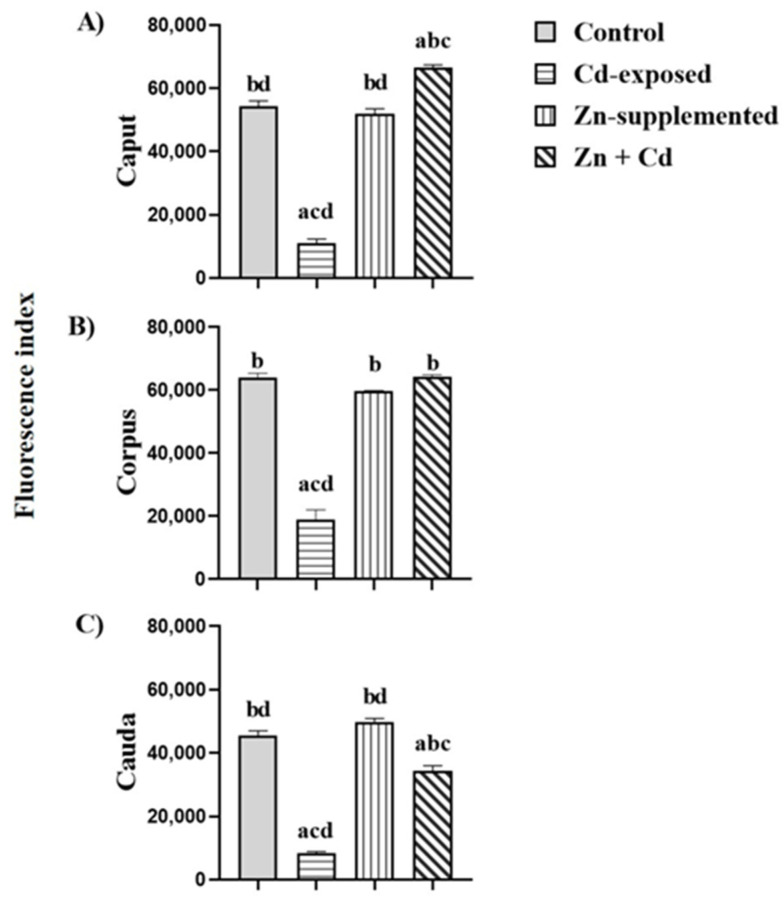
Fluorescence index of N-acetylglucosamine and/or sialic acid in the sperm of the three epididymal regions. Cd-exposed group decreased the presence of N-acetylglucosamine and/or sialic acid. However, in the Zn + Cd- group, Zn supplementation favored the presence of carbohydrates. Data expressed as means ± SEM for all analyzed groups (*n* = 7/treatment group). ^a^ *p* < 0.05 indicates differences vs. the control group; ^b^ *p* < 0.05 indicates differences vs. the Cd-exposed group; ^c^ *p* < 0.05 indicates differences vs. the Zn-supplemented group; ^d^ *p* < 0.05 indicates differences vs. the Zn + Cd group.

**Figure 8 ijms-26-04589-f008:**
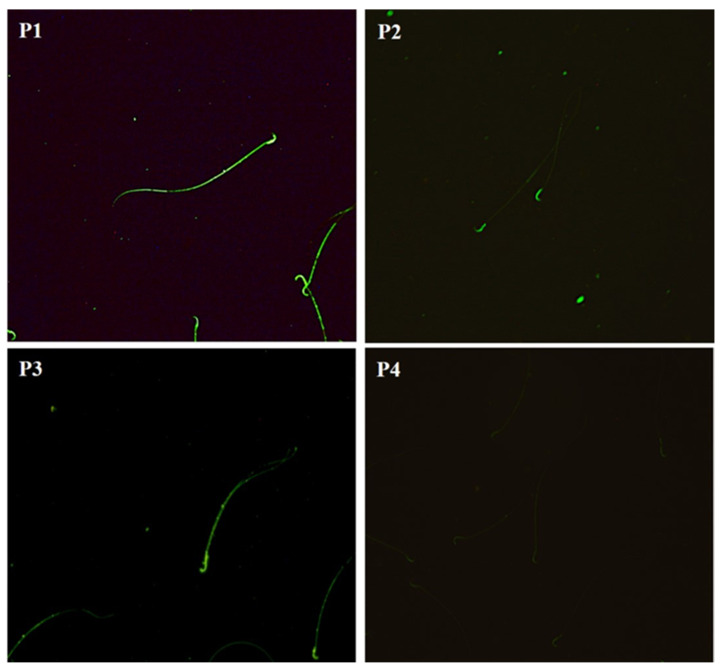
Micrographs of the distribution patterns of mannose fluorescence in sperm. The sperm showed different fluorescence patterns along their plasma membrane. Pattern 1 (P1): fluorescence in the head and main body; Pattern 2 (P2): fluorescence in the head; Pattern 3 (P3): fluorescence in the head and intermediate body; Pattern 4 (P4): sperm without fluorescence. Fluorescence microscopy, 20×.

**Figure 9 ijms-26-04589-f009:**
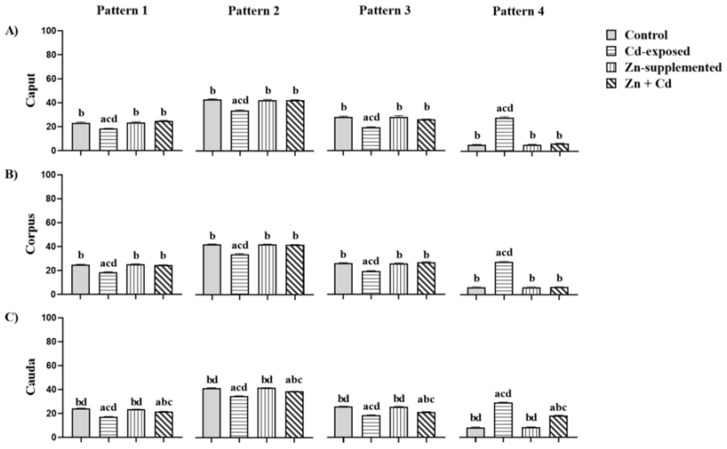
Percentages of distribution patterns of mannose fluorescence in the sperm membrane in the three epididymal regions. The fluorescence distribution of Pattern 2 was predominant in the epididymal regions, and between treatments, the group treated with Cd decreased the distribution of Patterns 1, 2, and 3 and increased Pattern 4. However, in the Zn + Cd group, Zn supplementation favored the distribution of mannose. Data expressed as means ± SEM for all analyzed groups (*n* = 7/treatment group). ^a^ *p* < 0.05 indicates differences vs. the control group; ^b^ *p* < 0.05 indicates differences vs. the Cd group; ^c^ *p* < 0.05 indicates differences vs. the Zn-supplemented group; ^d^ *p* < 0.05 indicates differences vs. the Zn + Cd group.

**Figure 10 ijms-26-04589-f010:**
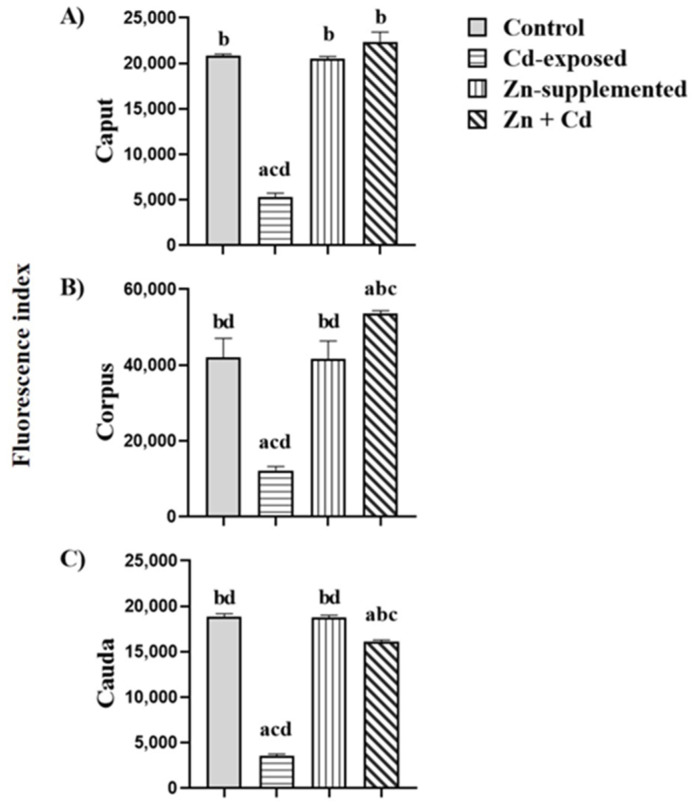
Fluorescence index of mannose in the sperm of the three epididymal regions. The Cd-exposed group decreased the presence of mannose. However, in the Zn + Cd group, Zn supplementation favored the presence of the carbohydrate. Data expressed as means ± SEM for all analyzed groups (*n* = 7/treatment group). ^a^ *p* < 0.05 indicates differences vs. the control group; ^b^ *p* < 0.05 indicates differences vs. the Cd-exposed group; ^c^ *p* < 0.05 indicates differences vs. the Zn-supplemented group; ^d^ *p* < 0.05 indicates differences vs. the Zn + Cd group.

**Figure 11 ijms-26-04589-f011:**
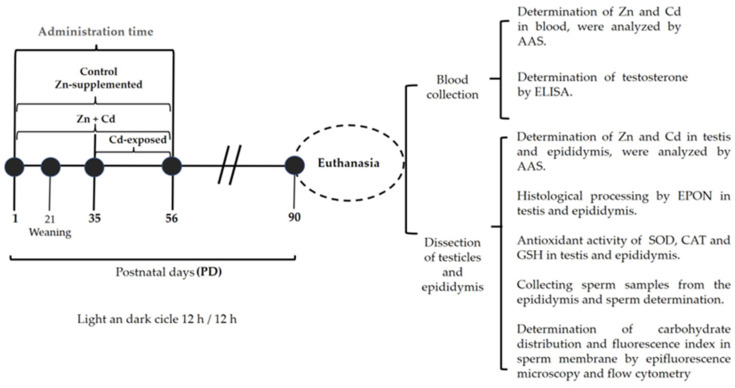
Experimental design and sample processing. Four experimental groups of Wistar rats were established: control (saline), Cd-exposed (CdCl_2_ 0.5 mg/kg body weight), Zn-supplemented (ZnCl_2_ 1 mg/kg body weight), and Zn + Cd (CdCl_2_ + ZnCl_2_). The treatments were administered intraperitoneally on the indicated days. After 90 days, the animals were euthanized to collect blood, testis, and epididymis samples. The following were analyzed: Cd and Zn metals via AAS; testosterone concentration via ELISA; histological processing via EPON; oxidative stress via ELISA (SOD, CAT, and GSH); sperm parameters and carbohydrates via epifluorescence and flow cytometry. Body weight (BW), postnatal day (PD), intraperitoneal (IP), atomic absorption spectrophotometry (AAS), enzyme-linked immunosorbent assay (ELISA), epoxy resin (EPON), superoxide dismutase (SOD), catalase (CAT), glutathione (GSH).

**Table 1 ijms-26-04589-t001:** Concentration of Cd and Zn in blood, testes, and epididymis.

	Control	Cd-Exposed	Zn-Supplemented	Zn + Cd
Blood (μg/mL)				
Cd	0.0048 ± 0.00078 ^b^	0.075 ± 0.031 ^acd^	0.0034 ± 0.0008 ^b^	0.0079 ± 0.0016 ^b^
Zn	0.66 ± 0.052 ^b^	0.41 ± 0.023 ^acd^	0.83 ± 0.067 ^b^	0.71 ± 0.059 ^b^
Testes (μg/g)				
Cd	0.0061 ± 0.0011 ^bd^	1.2 ± 0.13 ^acd^	0.0028 ± 0.0047 ^bd^	0.62 ± 0.052 ^abc^
Zn	0.96 ± 0.016 ^bc^	0.30 ± 0.0052 ^acd^	1.7 ± 0.30 ^abd^	1.1 ± 0.11 ^bc^
Epididymis (caput)				
Cd	0.0064 ± 0.00094 ^bd^	0.97 ± 0.092 ^acd^	0.0045 ± 0.012 ^bd^	0.59 ± 0.061 ^abc^
Zn	1.3 ± 0.17 ^b^	0.44 ± 0.044 ^acd^	1.8 ± 0.22 ^b^	1.5 ± 0.26 ^b^
Epididymis (cauda)				
Cd	0.0050 ± 0.00070 ^bd^	1.2 ± 0.063 ^acd^	0.0041 ± 0.0081 ^bd^	0.44 ± 0.045 ^abc^
Zn	1.3 ± 0.29 ^b^	0.30 ± 0.0036 ^acd^	1.8 ± 0.15 ^bd^	1.0 ± 0.022 ^bc^

Data are expressed as means ± SEM for all analyzed groups (*n* = 7/treatment group). ^a^ *p* < 0.05 indicates differences vs. the control group; ^b^ *p* < 0.05 indicates differences vs. the Cd-exposed group; ^c^ *p* < 0.05 indicates differences vs. the Zn-supplemented group; ^d^ *p* < 0.05 indicates differences vs. the Zn + Cd group.

**Table 2 ijms-26-04589-t002:** Serum testosterone concentration.

Control	Cd-Exposed	Zn-Supplemented	Zn + Cd
2.9 ± 0.098 ^b^	1.9 ± 0.28 ^acd^	3.1 ± 0.096 ^b^	2.6 ± 0.050 ^b^

Testosterone concentrations are expressed in ng/mL. Data are expressed as means ± SEM for all analyzed groups (*n* = 7/treatment group). ^a^ *p* < 0.05 indicates differences vs. the control group; ^b^ *p* <0.05 indicates differences vs. the Cd-exposed group; ^c^ *p* < 0.05 indicates differences vs. the Zn-supplemented group; ^d^ *p* < 0.05 indicates differences vs. the Zn + Cd group.

**Table 3 ijms-26-04589-t003:** Seminiferous tubule morphometric analysis.

	Control	Cd-Exposed	Zn-Supplemented	Zn + Cd
Diameter (µm)	331 ± 8.9	326 ± 6.2	336 ± 5.2	329 ± 7.6
Area (µm^2^)	63,212 ± 2330	61,124 ± 3300	71,766 ± 3064	67,190 ± 2344
Maturation index	9.6 ± 0.15	9.2 ± 0.21	9.7 ± 0.14	9.4 ± 0.19
Histopathological index	0.71 ± 0.18	0.86 ± 0.26	0.57 ± 0.20	0.71 ± 0.29

Data expressed as means ± SEM for all groups analyzed (*n* = 7/treatment group).

**Table 4 ijms-26-04589-t004:** Epididymal epithelium morphometric analysis.

		Control	Cd-Exposed	Zn-Supplemented	Zn + Cd
Length (µm)	caput	31 ± 0.30 ^b^	37 ± 0.62 ^acd^	30 ± 0.14 ^b^	29 ± 0.66 ^b^
	corpus	32 ± 0.33 ^b^	37 ± 0.80 ^acd^	31 ± 0.47 ^b^	30 ± 0.26 ^b^
	cauda	14 ± 0.30 ^bd^	35 ± 1.2 ^acd^	16 ± 0.36 ^bd^	21 ± 0.46 ^abc^
Length (µm)	caput	323 ± 16 ^b^	445 ± 12 ^acd^	306 ± 19 ^b^	330 ± 16 ^b^
	corpus	329 ± 10 ^b^	451 ± 16 ^acd^	334 ± 12 ^b^	328 ± 14 ^b^
	cauda	126 ± 5.5 ^bd^	281 ± 6.5 ^acd^	120 ± 5.1 ^bd^	202 ± 9.3 ^abc^

Data are expressed as means ± SEM for all analyzed groups (*n* = 7/treatment group). ^a^ *p* < 0.05 indicates differences vs. the control group; ^b^ *p* < 0.05 indicates differences vs. the Cd-exposed group; ^c^ *p* < 0.05 indicates differences vs. the Zn-supplemented group; ^d^ *p* < 0.05 indicates differences vs. the Zn + Cd group.

**Table 5 ijms-26-04589-t005:** Sperm parameters of epididymal regions.

		Control	Cd-Exposed	Zn-Supplemented	Zn + Cd
Concentration (×10^6^/mL)	Caput	69 ± 1.9 ^b^	43 ± 0.77 ^acd^	72 ± 2.2 ^bd^	64 ± 0.35 ^bc^
	Corpus	53 ± 0.70 ^b^	35 ± 1.8 ^acd^	56 ± 1.5 ^bd^	50 ± 0.49 ^bc^
	Cauda	87 ± 1.1 ^bd^	51 ± 0.43 ^acd^	88 ± 1.0 ^bd^	82 ± 0.22 ^abc^
Viability (%)	Caput	94± 0.42 ^b^	83 ± 1.0 ^acd^	94 ± 0.73 ^b^	92 ± 0.48 ^b^
	Corpus	95 ± 0.86 ^b^	78 ± 1.5 ^acd^	96 ± 0.33 ^bd^	92 ± 0.37 ^bc^
	Cauda	94 ± 0.43 ^bd^	75 ± 1.8 ^acd^	93 ± 0.58 ^bd^	88 ± 0.84 ^abc^
Morphological Normality (%)	Caput	95 ± 0.56 ^b^	72 ± 0.86 ^acd^	94 ± 0.49 ^b^	94 ± 0.42 ^b^
	Corpus	97 ± 0.22 ^b^	73 ± 0.92 ^acd^	97 ± 0.40 ^b^	96 ± 0.40 ^b^
	Cauda	94 ± 0.26 ^bd^	70 ± 1.2 ^acd^	94 ± 0.54 ^bd^	90 ± 0.42 ^abc^

Data are expressed as means ± SEM for all groups analyzed (*n* = 7/treatment group). ^a^ *p* < 0.05 indicates differences vs. the control group; ^b^
*p* < 0.05 indicates differences vs. the Cd-exposed group; ^c^ *p* < 0.05 indicates differences vs. the Zn-supplemented group; ^d^ *p* < 0.05 indicates differences vs. the Zn + Cd group.

## Data Availability

The original contributions presented in this study are included in the article. Further inquiries can be directed to the corresponding author.

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
