# Peer review of "Early Zinc Supplementation Enhances Epididymal Sperm Glycosylation, Endocrine Activity, and Antioxidant Activity in Rats Exposed to Cadmium"

_ijms, 2025, doi:10.3390/ijms26104589_

Round 1

Reviewer 1 Report

Comments and Suggestions for Authors

The manuscript is well-structured and clearly organized. However, some sections, especially the Introduction and Discussion, could be further streamlined for better readability. For instance, consolidating similar findings and shortening repetitive discussions would enhance the flow of the text.

In the study, ANOVA and Tukey's test were employed for data analysis, but it was not stated whether the normality of data and homogeneity of variance had been examined. It is recommended to supplement relevant statistical test results and explain how non-normally distributed data were handled.
The rationale for selecting the doses of Cd and Zn (0.5 mg/kg and 1 mg/kg) and treatment duration (35-56 days) was not fully elucidated. It is suggested to cite relevant literature or pre-experimental data to demonstrate the scientific validity of the chosen doses and time window.
Although the evaluation methods for sperm concentration, motility, and morphology were based on WHO standards, the specific operational procedures and scoring criteria were not provided. It is recommended to supplement detailed operational procedures to ensure consistency and objectivity in the evaluation.
The discussion section emphasized the protective effect of Zn against Cd toxicity but did not fully explore the possible molecular mechanisms (such as competitive binding between Zn and Cd, specific regulation of antioxidant pathways, etc.). It is suggested to conduct an in-depth analysis of the mechanisms by integrating existing literature.

Author Response

Response to Reviewer's comments

Reviewer #1 The manuscript is well-structured and clearly organized. However, some sections, especially the Introduction and Discussion, could be further streamlined for better readability. For instance, consolidating similar findings and shortening repetitive discussions would enhance the flow of the text.
Response: We appreciate the thoughtful suggestion from the reviewer. We reviewed carefully ans reworded paragraphs and sentences to enhance clarity and conciseness throughout the manuscript.

1) In the study, ANOVA and Tukey's test were employed for data analysis, but it was not stated whether the normality of data and homogeneity of variance had been examined. It is recommended to supplement relevant statistical test results and explain how non-normally distributed data were handled.
Response: We thank the reviewer for their observation. We agree with their comment; the correct normality test is Shapiro-Wilk. This is because the n for each group is 7 for each parameter analyzed. The following has been added to the statistical analysis section (L 852-859):

All data are presented as mean ± standard error of the mean (SEM). The normality of the data distribution was assessed using the Shapiro-Wilk test. For comparisons between four groups (Ctrl vs Cd vs Zn vs Zn + Cd), one-way ANOVA was used, followed by Tukey's test for multiple comparisons. A p-value < 0.05 was considered statistically significant. All analyses were performed using GraphPad Prism version 8.01.

2) The rationale for selecting the doses of Cd and Zn (0.5 mg/kg and 1 mg/kg) and treatment duration (35-56 days) was not fully elucidated. It is suggested to cite relevant literature or pre-experimental data to demonstrate the scientific validity of the chosen doses and time window.
Response: We thank the reviewer for their comment. In the M&M section (4.2 Treatments). The justification for the choice of doses, the time interval chosen, and the bibliography have been added (L 676-681):

Doses and administration were chosen based on previous studies [Li et al., 2017; Babaknejad et al., 2018; Hernández-Rodríguez et al., 2021a, b). The administration of Zn from the day of birth is proposed as a preventive strategy to assess its potential protective effect against Cd exposure, which includes 35 to 56 postnatal days. This period is crucial for reproductive development as it coincides with the increase in testosterone levels, the initiation of the first wave of spermatogenesis and the maturation of the epididymal epithelium.

Citations:

  • Li, Z., Li, Y., Zhou, X., Cao, Y., & Li, C. (2017). Preventive effects of supplemental dietary zinc on heat-induced damage in the epididymis of boars. Journal of thermal biology, 64, 58-66. https://doi.org/10.1016/j.jtherbio.2017.01.002
  • Babaknejad, N., Bahrami, S., Moshtaghie, A. A., Nayeri, H., Rajabi, P., & Iranpour, F. G. (2018). Cadmium Testicular Toxicity in Male Wistar Rats: Protective Roles of Zinc and Magnesium. Biological trace element research, 185(1), 106-115. https://doi.org/10.1007/s12011-017-1218-5. https://doi.org/10.1007/s12011-017-1218-5
  • Hernández-Rodríguez, J., Arenas-Ríos, E., Jiménez-Morales, I., Cortés-Barberena, E., Montes, S., Vigueras-Villaseñor, R. M., & Arteaga-Silva, M. (2021a). Postnatal cadmium administration affects the presence and distribution of carbohydrates in the sperm membrane during maturation in the epididymis in adult Wistar rats. Reproduction, fertility, and development, 33(5), 349-362. https://doi.org/10.1071/RD20167
  • Hernández-Rodríguez, J., López, A. L., Montes, S., Bonilla-Jaime, H., Morales, I., Limón-Morales, O., Ríos, C., Hernández-González, M., Vigueras-Villaseñor, R. M., & Arteaga-Silva, M. (2021b). Delay in puberty indices of Wistar rats caused by Cadmium. Focus on the redox system in reproductive organs. Reproductive toxicology (Elmsford, N.Y.), 99, 71-79. https://doi.org/10.1016/j.reprotox.2020.11.010

3) Although the evaluation methods for sperm concentration, motility, and morphology were based on WHO standards, the specific operational procedures and scoring criteria were not provided. It is recommended to supplement detailed operational procedures to ensure consistency and objectivity in the evaluation.
Response: We thank the reviewer for the comment. operating procedures are supplemented in the M&M section (4.4.5. Collecting sperm samples from the epididymis and sperm determination) (L 786-819):

4.4.5.1. Sperm concentration
A 1:50 v/v dilution was performed in Eppendorf tubes of the spermatozoa obtained and washed with distilled water (980 μL of distilled water and 20 μL of the sperm sample were placed). For counting, a Neubauer chamber was used where an aliquot of 10 μL of the sperm-water dilution (1:50 dilution) was placed on each side, allowed to stand for 1 minute. Counting was performed on both sides of the chamber using an Olympus BX 51 optical microscope (Tokyo, Japan). The number of spermatozoa present in the first upper left frame of the chamber was counted. According to the number of spermatozoa counted in the first upper frame were the frames to be counted from the Neubauer chamber. Five frames were counted from each side of the chamber. The sperm concentration was expressed in millions/mL, for which the mean of the count of both sides of the chamber was obtained and divided by the corresponding conversion factor.

4.4.5.2. Sperm viability
5 μL of eosin-nigrosin solution was placed on a slide and 5 μL of the sperm sample. They were mixed and then smeared along the slide, allowed to dry at a temperature of 36.5 °C on a thermoplate. It was observed under an Olympus BX 51 optical microscope (Tokyo, Japan), at 40X and the number of stained (dead) or unstained (live) cells was counted with a laboratory counter. 100 spermatozoa were counted per sample, and the number of live spermatozoa was represented in percentage of vitality.

4.4.5.3. Sperm morphology
An aliquot of 10 μL of the sperm sample was taken, placed on a slide and smeared along the slide. Samples were allowed to dry for 5 minutes at room temperature, then stained using a rapid staining kit Eperma-form ESF-076, FertiMexico). The samples were placed in a fixative and then in the dyes according to the specifications of the kit. Subsequently, the samples were observed under an Olympus BX 51 optical microscope (Tokyo, Japan) at 40X, and were evaluated under the following criteria: When the spermatozoa presented a hooked head, they were considered as spermatozoa with normal morphology. If the spermatozoa head was observed with an encircled, pyriform, vacuolated, double head or combinations of these, it was considered as abnormal morphology. As for the intermediate piece, it was considered a normal morphology when it presented thinness, and represented the third part of the width of the head and was aligned with the longitudinal axis of the head. For the flagellum, a normal morphology was considered when it did not present cytoplasmic drop, folding, breakage, hairpin formation or coiling.

4) The discussion section emphasized the protective effect of Zn against Cd toxicity but did not fully explore the possible molecular mechanisms (such as competitive binding between Zn and Cd, specific regulation of antioxidant pathways, etc.). It is suggested to conduct an in-depth analysis of the mechanisms by integrating existing literature.
Response: We appreciate the reviewer's comments and have considered complementing, the information with respect to the molecular mechanisms between Zn and Cd, which have been appended in the introduction and discussion section (L 429-433, 547-573).

Reviewer 2 Report

Comments and Suggestions for Authors

In this study attempts were made to investigate whether Zinc (Zn), Cadmium (Cd) or both Zn+Cd could protect or enhance glycosylation changes in rat sperm membrane. In this study it seems that the rats were not sexually matured throughout the experimental period (75-100 days). Why sexually mature rats were not considered in this study? If spermatogenesis lasts for approximately 54 days in the rats, would Cd treatment had any impact on the cell types, as indicated in Figures 1-2 (L459-461). The Reviewer suggests that the following comments will be helpful to improve the quality of the manuscript.

Comments are as follows:

1. This study has more than one objectives (L115-120), and the present one is incomplete. "glycosylation of membrane of sperm….’’ Sperm from where – testis, or epididymis? What about testosterone measurements, morphology and antioxidant analyses?

2. Why attempts were not made to measure the activities of the sperm-associated glycosyltransferases (L647-648/L659-660), as indicated, these enzymes catalyze the transfer of sugar residues from nucleotide sugar donor to the sugar chains on glycoproteins and glycolipids of sperm and fluid from the different regions of the epididymis (Tulsiani et al., 1993). Perhaps such analysis would have provided more information about the post-translational glycosylation changes in the sperm during epididymal maturation (L609-612).

 3.Results

a) No mention about the weight of the animals, or their reproductive organs in the M & M (L123-132).

b) Re-check the statistical analysis, L811-816 (all the tables and Figures 4, 6-, 9-10). The Tukey multiple comparison test compares all the means simultaneously, including the control and the treated groups (that is, the control has to show the significant difference, L814?), while the Dunnett’s multiple comparison test compares the control group against the treated groups (the control group does not show the significant difference, as indicated in this study-L814?).

c) Following re-analysis of the statistical test, describe the results of the above-mentioned tables and figures.

d) Figures 5 and 8 are unclear. Figure 11 needs to be improved because it is difficult to follow.

4. M & M

a) Rats are susceptible to physical stress, as in this study the daily injection of the Zn/Cd solutions. That is why it would been appropriate that, besides the saline treated control group, an non-injected animal group should have been considered as the control (that is, a control without no injection and the vehicle control group).

b) Was Cd dissolved in the same saline solution used for the control?

5. Should consider to shorten the Discussion. The biochemical roles of zinc and its ligands in sperm functions, and sperm egg-related fertilization processes have been well documented in numerous studies.

Author Response

Reviewer #2
In this study attempts were made to investigate whether Zinc (Zn), Cadmium (Cd) or both Zn+Cd could protect or enhance glycosylation changes in rat sperm membrane. In this study it seems that the rats were not sexually matured throughout the experimental period (75-100 days). Why sexually mature rats were not considered in this study? If spermatogenesis lasts for approximately 54 days in the rats, would Cd treatment have any impact on the cell types, as indicated in Figures 1-2 (L459-461). The Reviewer suggests that the following comments will be helpful to improve the quality of the manuscript.

Response to Reviewer's comments
We appreciate the thoughtful suggestion from the reviewer. We reviewed carfully ans reworded paragraphs and sentences to enhance clarity and conciseness throughout the manuscript.

Response: We thank the reviewer for his comments. In the present study, Zn administration from the day of birth to 56 postnatal life was performed as a protective strategy to Cd exposure. In the postnatal ages from 35 to 56, which is a crucial period for the reproductive development of our study model, the Wistar rat, as it has been reported to coincide with the increase in testosterone concentration, with the onset of the first wave of spermatogenesis and maturation of the epididymal epithelium (Picut et al., 2018). Regarding glycosylation, it has been described that it can be affected since spermatogenesis (Tecle & Gagneux et al., 2015), moreover, as reported, any damage caused by Cd in the testis can affect the function of the epididymis (De Grava Kempinas & Klinefelter, 2015; Machado Neves, 2022). However, in our study, exposure to Cd in the seminiferous epithelium did not cause structural damage, but it did reduce testosterone concentration, an alteration that is reflected in the function of the epididymis when evaluating glycosylation in the plasma membrane of spermatozoa. As described, the epididymis is more sensitive to damage by heavy metals, affecting cell development, differentiation and expansion (Machado Neves, 2022). Regarding the period or stage of the subjects used in this study, it has been reported by other groups and in our laboratory the effects of Cd in the different life stages of the rat (neonatal, infant, juvenile and adult), finding alterations at the testicular (Shi and Fu, 2019; Bhardwaj et al., 2021) and epididymal (Hernández-Rodríguez et al., 2021) levels, even in adult animals it has been reported the alteration of male sexual behavior (Arteaga Silva et al., 2023). Thus, our interest in this study was to see the protective effect of Zn at the same stages where the adverse effects of Cd have been reported.

Citations:

  • Picut, C. A., Ziejewski, M. K., & Stanislaus, D. (2018). Comparative Aspects of Pre- and Postnatal Development of the Male Reproductive System. Birth defects research, 110(3), 190-227. https://doi.org/10.1002/bdr2.1133Tecle, E., & Gagneux, P. (2015). Sugar-coated sperm: Unraveling the functions of the mammalian sperm glycocalyx. Molecular reproduction and development, 82(9), 635-650. https://doi.org/10.1002/mrd.22500De Grava Kempinas, W., & Klinefelter, G. R. (2015). Interpreting histopathology in the epididymis. Spermatogenesis, 4(2),e979114. https://doi.org/10.4161/21565562.2014.979114
  • Machado-Neves M. (2022). Effect of heavy metals on epididymal morphology and function: An integrative review. Chemosphere, 291(Pt 2), 133020. https://doi.org/10.1016/j.chemosphere.2021.133020
  • Shi, X., & Fu, L. (2019). Piceatannol inhibits oxidative stress through modification of Nrf2-signaling pathway in testes and attenuates spermatogenesis and steroidogenesis in rats exposed to cadmium during adulthood. Drug Design, Development and Therapy, 13, 2811-2824. https://doi.org/10.2147/DDDT.S198444
  • Bhardwaj, J. K., Siwach, A., Sachdeva, D., & Sachdeva, S. N. (2024). Revisiting cadmium-induced toxicity in the male reproductive system: an update. Archives of toxicology, 98(11), 3619-3639. https://doi.org/10.1007/s00204-024-03871-7.
  • Hernández-Rodríguez, J., López, A. L., Montes, S., Bonilla-Jaime, H., Morales, I., Limón-Morales, O., Ríos, C., Hernández-González, M., Vigueras-Villaseñor, R. M., & Arteaga-Silva, M. (2021). Delay in puberty indices of Wistar rats caused by Cadmium. Focus on the redox system in reproductive organs. Reproductive toxicology (Elmsford, N.Y.), 99, 71-79. https://doi.org/10.1016/j.reprotox.2020.11.010
  • Arteaga-Silva, M., Limón-Morales, O., Bonilla-Jaime, H., Vigueras-Villaseñor, R. M., Rojas-Castañeda, J., Hernández-Rodríguez, J., Montes, S., Hernández-González, M., & Ríos, C. (2023). Effects of postnatal exposure to cadmium on male sexual incentive motivation and copulatory behavior: Estrogen and androgen receptors expression in adult brain rat. Reproductive toxicology (Elmsford, N.Y.), 120, 108445. https://doi.org/10.1016/j.reprotox.2023.108445.

2) This study has more than one objective (L115-120), and the present one is incomplete. "glycosylation of membrane of sperm....'' Sperm from where - testis, or epididymis? What about testosterone measurements, morphology and antioxidant analyses?
Response: We appreciate the reviewer's comments. We agree to adjust the title of our work, adding the source of the sperm (epididymis), the endocrine evaluation, and the antioxidant activity. In addition, the objective of the study has been adjusted as follows (L 2-3, 119-124):

Title
Early Zinc Supplementation Enhances Epididymal Sperm Glycosylation, Endocrine and Antioxidant Activity in Rats Exposed to Cadmium

Objective
Therefore, this study aimed to analyze and determine whether Zn pretreatment can protect and enhance the glycosylation of the epididymal sperm membrane, endocrine and antioxidant activity when exposed to Cd.

3) Why attempts were not made to measure the activities of the sperm-associated glycosyltransferases (L647-648/L659-660), as indicated, these enzymes catalyze the transfer of sugar residues from nucleotide sugar donor to the sugar chains on glycoproteins and glycolipids of sperm and fluid from the different regions of the epididymis (Tulsiani et al., 1993). Perhaps such analysis would have provided more information about the post-translational glycosylation changes in the sperm during epididymal maturation (L609-612).
Response: We appreciate the reviewer's comments. We have indeed considered that measuring the activity of glycan-modifying enzymes (glycosyltransferases and glycosidases) would provide us with more information on the positive effects of Zn. However, this is an objective that we plan to analyze in the next study, thereby complementing our line of research.

Results
a) No mention about the weight of the animals, or their reproductive organs in the M & M (L123-132).
Response: We appreciate the reviewer's comments. The weights of the animals and their organs (testes and epididymides) have been added to the M&M section (4.3. Experimental procedure (L 684,691):

At 90 days of age, animals were weighed and euthanized by decapitation under anesthesia…The epididymides and testes were dissected bilaterally and weighed……

b) Re-check the statistical analysis, L811-816 (all the tables and Figures 4, 6-, 9-10). The Tukey multiple comparison test compares all the means simultaneously, including the control and the treated groups (that is, the control has to show the significant difference, L814?), while the Dunnett's multiple comparison test compares the control group against the treated groups (the control group does not show the significant difference, as indicated in this study-L814?).

c) Following re-analysis of the statistical test, describe the results of the above-mentioned tables and figures.
Response: We appreciate your corrections and suggestions to the reviewer.

Regarding the question; the control has to show the significant difference?

In our study it was not necessary to place the differences in the control group, which is the reference group to compare with the groups with different treatments.

4.4.9. Statistical Analysis (L 851-858):
We did not use Dunnet's statistical test that compares the differences of the control group vs. the treatments. Because the interest of the study is the multiple comparisons that Tukey's post hoc test gives us by comparing the means of the control group and the different groups simultaneously. Thus, the differences considered were the control group vs Cd vs Zn vs Zn vs Zn + Cd and the comparison between the groups Zn vs Cd, Zn vs Zn + Cd and Cd vs Zn + Cd, for each parameter analyzed. p < 0.05 was considered statistically significant. All analyses were performed using GraphPad Prism version 8.01.

d) Figures 5 and 8 are unclear. Figure 11 needs to be improved because it is difficult to follow.
Response: We thank the reviewer for his comments. Figures 5, 8 and 11 were adjusted for clarity and understanding.

M & M
a) Rats are susceptible to physical stress, as in this study the daily injection of the Zn/Cd solutions. That is why it would have been appropriate that, besides the saline treated control group, an non-injected animal group should have been considered as the control (that is, a control without no injection and the vehicle control group).
Response: We appreciate the reviewer's comments regarding this point in our laboratory, previously it has been verified that the injection does not cause stress in the individual that is reflected in alterations for each of the parameters analyzed, in addition, the control group has the same experimental manipulation as the treated groups, (housed under the same biotherium conditions), this has been reported in previous works (Hernández Rodríguez et al., 2021; Arteaga-Silva et al., 2023).

Citations:

  • Hernández-Rodríguez, J., López, A. L., Montes, S., Bonilla-Jaime, H., Morales, I., Limón-Morales, O., Ríos, C., Hernández-González, M., Vigueras-Villaseñor, R. M., & Arteaga-Silva, M. (2021b). Delay in puberty indices of Wistar rats caused by Cadmium. Focus on the redox system in reproductive organs. Reproductive toxicology (Elmsford, N.Y.), 99, 71-79. https://doi.org/10.1016/j.reprotox.2020.11.010
  • Arteaga-Silva, M., Limón-Morales, O., Bonilla-Jaime, H., Vigueras-Villaseñor, R. M., Rojas-Castañeda, J., Hernández-Rodríguez, J., Montes, S., Hernández-González, M., & Ríos, C. (2023). Effects of postnatal exposure to cadmium on male sexual incentive motivation and copulatory behavior: Estrogen and androgen receptors expression in adult brain rat. Reproductive toxicology (Elmsford, N.Y.), 120, 108445. https://doi.org/10.1016/j.reprotox.2023.108445.

b) Was Cd dissolved in the same saline solution used for the control?
Response: We thank the reviewer for his comments. Both CdCl2 and ZnCl2 solutions were dissolved equally with saline administered to the control group. The information was added in the M&M section (4.2 Treatments (L 669-673):

The first group was used as control and received 100 μL vehicle (saline solution), the second group received 0.5 mg/kg/100µL of body weight (bw) of Cd (CdCl2, Sigma-Aldrich, St. Louis, MO) diluted in saline solution, the third group received 1 mg/kg/100µL of bw of Zn (ZnCl2, Sigma Aldrich, Mexico), diluted in saline solution, and the fourth group received Zn + Cd at the respective doses.

c) Should consider to shorten the Discussion. The biochemical roles of zinc and its ligands in sperm functions, and sperm egg-related fertilization processes have been well documented in numerous studies.
Response: We thank the reviewer for his comments. The discussion was adapted to improve the flow of the text..

Round 2

Reviewer 2 Report

Comments and Suggestions for Authors

The Authors have addressed my concerns, except for the comments regarding the statistical analysis.

b) Re-check the statistical analysis, L811-816 (all the tables and Figures 4, 6-, 9-10). The Tukey multiple comparison test compares all the means simultaneously, including the control and the treated groups (that is, the control has to show the significant difference, L814?), while the Dunnett's multiple comparison test compares the control group against the treated groups (the control group does not show the significant difference, as indicated in this study-L814?).

Response: In our study it was not necessary to place the differences in the control group, which is the reference group to compare with the groups with different treatments.

The Reviewer does not agree with the Authors’ reply. For the Tukey multiple comparison test (is the same as the Tukey’s post hoc test) the significant differences should be inserted above the column bars for all the parameters analyzed by the test (including the control), even though the differences are nonsignificant. As the regard the Dunnett's multiple comparison test the significant differences are  not indicated for the control. That’s the principle of both tests.

Author Response

Reviewer #2 The Authors have addressed my concerns, except for the comments regarding the statistical analysis.

The Reviewer does not agree with the Authors’ reply. For the Tukey multiple comparison test (is the same as the Tukey’s post hoc test) the significant differences should be inserted above the column bars for all the parameters analyzed by the test (including the control), even though the differences are nonsignificant. As the regard the Dunnett's multiple comparison test the significant differences are not indicated for the control. That’s the principle of both tests.

Response: We appreciate the reviewer’s comments and feedback.

Significant differences were inserted into the tables and figures as suggested for the control group. Additionally, we added a letter (d) to the Zn + Cd-treated group to account for the differences with the control group. Additionally, non-significant differences (NS) with respect to the control group were inserted.